# Interferon restores replication fork stability and cell viability in BRCA-defective cells via ISG15

Ramona N. Moro [1,7], Uddipta Biswas[1,7], Suhas S. Kharat[2], Filip D. Duzanic[1], Prosun Das[3], Maria Stavrou [1], Maria C. Raso[1], Raimundo Freire [4,5,6], Arnab Ray Chaudhuri[3], Shyam K. Sharan [2] & Lorenza Penengo [1] ✉

DNA replication and repair defects or genotoxic treatments trigger interferon (IFN)-mediated inflammatory responses. However, whether and how IFN signaling in turn impacts the DNA replication process has remained elusive. Here we show that basal levels of the IFN-stimulated gene 15, ISG15, and its conjugation (ISGylation) are essential to protect nascent DNA from degradation. Moreover, IFNβ treatment restores replication fork stability in BRCA1/2-deficient cells, which strictly depends on topoisomerase-1, and rescues lethality of BRCA2-deficient mouse embryonic stem cells. Although IFNβ activates hundreds of genes, these effects are specifically mediated by ISG15 and ISGylation, as their inactivation suppresses the impact of IFNβ on DNA replication. ISG15 depletion significantly reduces cell proliferation rates in human BRCA1-mutated triple-negative, whereas its upregulation results in increased resistance to the chemotherapeutic drug cisplatin in mouse BRCA2-deficient breast cancer cells, respectively. Accordingly, cells carrying BRCA1/2 defects consistently show increased ISG15 levels, which we propose as an in-built mechanism of drug resistance linked to BRCAness.

The fidelity of genome duplication is crucial for genome maintenance. A variety of stresses can challenge this fundamental process resulting in "replication stress," characterized by alteration of the rate and the fidelity of DNA synthesis. If not timely and properly addressed, replication stress can lead to replication fork collapse and DNA double-strand breaks, promoting cancer development. Multiple mechanisms and factors regulating DNA replication fork progression and stability cooperate to ensure genome integrity upon replication stress. In fact, germline or acquired mutations targeting these factors represent a relevant percentage of alterations observed in human malignancies. Among others, mutations in *BRCA1*/2 genes are associated with an increased risk of developing different types of tumors, including breast, ovarian, pancreatic and prostate cancer[1]. BRCA1 and BRCA2 are key proteins in repairing DNA breaks, by promoting the homologous recombination (HR) DNA repair pathway, and in maintaining the stability of newly synthesized DNA strands, by protecting stalled replication forks from degradation, hence preventing chromosomal aberrations[2–4].

A side effect of genetic defects or genotoxic treatments challenging replication fork stability is the generation of byproducts (ssDNA, oligonucleotides, DNA:RNA hybrids) and their accumulation in the cytosol. These nucleic acids mimic pathogen infection and are typical

[1]University of Zurich, Institute of Molecular Cancer Research, 8057 Zurich, Switzerland. [2]Mouse Cancer Genetics Program, Center for Cancer Research, National Cancer Institute, National Institutes of Health, Frederick 21702 MD, USA. [3]Department of Molecular Genetics, Erasmus MC Cancer Institute, Erasmus University Medical Center, 3015GD Rotterdam, the Netherlands. [4]Fundación Canaria del Instituto de Investigación Sanitaria de Canarias (FIISC), Unidad de Investigación, Hospital Universitario de Canarias, La Laguna, Santa Cruz de Tenerife, Spain. [5]Instituto de Tecnologías Biomédicas, Universidad de La Laguna, 38200 La Laguna, Spain. [6]Universidad Fernando Pessoa Canarias, Las Palmas de Gran Canaria, Spain. [7]These authors contributed equally: Ramona N. Moro, Uddipta Biswas. ✉e-mail: penengo@imcr.uzh.ch

activators of cyclic GMP-AMP synthase (cGAS), a DNA sensor that triggers innate immune responses through the production of the second messenger cyclic GMP-AMP (cGAMP) and activation of the adapter protein STING[5]. Downstream activation of TBK1 and IFN regulatory factor 3 (IRF3) then occurs, resulting in the activation of type I IFN and the upregulation of the IFN-stimulated genes, ISGs[6–8]. This inflammatory response, induced by chromosomal instability, has broad and diverse effects depending on the cancer type and context, as acute inflammation has immune-stimulatory effects against malignancies while chronic inflammation may promote cancer development[7,9–11], and can even be exploited for immunotherapeutic purposes[12], underlining the importance of better understanding the links between replication stress and the immune system.

The mechanisms through which genomic instability triggers the immune response have been extensively investigated in recent years. Yet, the consequences of this activation on DNA replication, cell fitness and homeostasis are far less understood. We recently reported that type I IFN (i.e., IFNβ), through the upregulation of ISG15, promotes deregulated DNA replication fork progression, representing the first natural event leading to an acceleration of DNA replication rate, with detrimental consequences for the cells[13]. ISG15 is a ubiquitin-like modifier that exerts its functions via covalent conjugation to targets— referred as ISGylation—by means of the E1 activating enzyme (UBE1L), the E2 conjugating enzyme (UBCH8) and the E3 ligases (HERC5, TRIM25 and HHARI). Interestingly, ISG15 also functions through non-covalent interactions with intracellular proteins and as a secreted molecule[14–16]. ISG15 plays a central role in the antimicrobial response by protecting the host during infection[17], but it is also frequently deregulated in cancer[18], yet its exact role is controversial.

Only recently we began to appreciate the role of the ISG15 system in the DNA damage response, DNA replication and genome stability. Upon UV irradiation, TRIM25 (also known as EFP) interacts with mono-ubiquitinated PCNA (proliferating cell nuclear antigen) and promotes its ISGylation, which in turn leads to the de-ubiquitination of PCNA and the termination of error-prone translesion DNA synthesis, thus limiting an excessive mutagenesis[19]. Later, we reported that ISG15 upregulation provokes accelerated and unrestrained DNA replication by promoting RECQ1-dependent restart of stalled forks, leading to accumulation of DNA damage and increased chromosomal aberrations[13]. More recently, it has been reported in mouse embryonic fibroblasts (MEFs) that ISG15 and ISGylation are strongly induced, and enriched at replication forks upon the inactivation of the MRN complex (MRE11, NBS1, and RAD50), a key component of the DNA damage response that is also associated with DNA replication forks[20]. Moreover, loss of ISG15 was linked to replication fork stalling, genomic aberrations, and drug sensitivity. Although this study revealed the presence of ISGylated proteins at the replication forks in *Nbs1*-deleted MEFs and implicated ISG15 in limiting replication stress, how ISGylation modulates DNA replication, and the mechanisms underlying this regulation are still poorly explored.

Here, we addressed the effects of IFNβ signaling – and ISG15 – on the stability of the DNA replication forks both in unperturbed cells and in pathological genetic contexts, such as the BRCA1/2-deficiencies, characterized by high genomic instability, DNA repair defects and severe replication stress. On one side we observed that basal levels of ISG15 and the ISGylating enzyme UBE1L are required to ensure proper replication fork progression and stability of nascent DNA. On the other, we found that treatment with low doses of IFNβ—which leads to increased ISG15 levels—completely restores replication fork stability in BRCA1/2-deficient cells. This effect, observed consistently across multiple human and mouse cell lines, including patient derived BRCA1/2-defective lines, is entirely dependent on ISG15 and on the enzymes mediating its covalent conjugation, namely UBE1L and TRIM25, and strictly requires the topoisomerase-1 TOP1. Remarkably, IFNβ treatment is also able to rescue the viability of BRCA2-deficient mouse embryonic stem cells (mESCs) in an ISG15/ISGylation-dependent

manner and the upregulation of ISG15 confers drug resistance to BRCA2-deficient cells. Consistent with this, loss of ISG15 has dramatic effects on cell fitness, impairing proliferation of BRCA1-mutated triple-negative breast cancer cells. Altogether, these findings reveal that the IFNβ/ISG15 system controls fork stability in clinically relevant pathological contexts, increasing the fitness of BRCA1/2-deficient cancer cells and affecting the drug response.

## Results

### ISG15 and its conjugation are required to ensure stability of nascent DNA

We recently reported that upregulation of ISG15 as well as low doses of IFNβ treatment accelerates DNA replication fork progression in many different cell types[13]. In the present study, we aimed to elucidate whether and how ISG15 and IFNβ control DNA replication fork integrity and which mechanisms underlie this regulation. First, using the DNA fiber spreading technique, in which ongoing DNA synthesis is labeled with halogenated thymidine analogs (CldU followed by IdU) that can be recognized by specific antibodies[21], we examined DNA replication upon loss of ISG15 and observed consistent impairment of replication fork progression and asymmetry of sister forks stemming from the same replication origin, compared to parental cells (Fig. 1a–c and Supplementary Fig. 1a–d). To better understand this effect, we employed a modified version of the DNA fiber assay to evaluate the stability of newly synthesized DNA upon fork stalling. To pause DNA replication, after labeling with CldU/IdU cells are treated with hydroxyurea (HU; 4 mM, 4 h), which depletes the cellular pool of dNTPs by inhibiting the ribonucleotide reductase, and induces fork stalling (Fig.1d). In control cells, the arrested forks are protected by several factors[22], and therefore no extensive degradation of stalled forks (i.e., reduction of the green tract) is observed, resulting in the ratio between IdU (green) and CldU (red) of approximately 1. Surprisingly, we found that loss of ISG15 in MEFs leads to significant reduction of the IdU/CldU ratio compared to ISG15-proficient cells, indicating marked degradation of the nascent DNA (Fig. 1e, f). Similar effects were observed in human osteosarcoma U2OS cells upon depletion of ISG15 (Fig. 1g, h). To further assess that the reduced ratio of IdU/CldU tract lengths upon HU is due to degradation of nascent DNA, we pre-treated cells with mirin—a specific inhibitor of the nuclease MRE11 that has been largely involved in the degradation of stalled forks in condition of replication stress[3,23–25]—and found that fork degradation is fully rescued (Fig. 1i–k). These results indicate that basal levels of ISG15 are required for the stability of nascent DNA and that its loss induces MRE11-dependent fork degradation of HU-stalled forks.

Being part of the ubiquitin family, an important mechanism of action of ISG15 is via covalent conjugation to target proteins. Hence, we aimed to investigate whether ISG15 conjugation is required to exert this new function in replication fork protection. Therefore, we analyzed fork stability in cells upon loss of UBE1L, the sole E1 activating enzyme able to promote the initial step of ISGylation, which is thus essential for ISG15 conjugation in cells. Interestingly, we found that UBE1L is necessary to maintain the stability of stalled forks both in MEFs and in U2OS (Fig. 1l–n). To further explore this point, we tested whether a conjugation-defective form of ISG15, referred to as ΔGG, which lacks the C-terminal GlyGly motif implicated in the covalent binding of ubiquitin and ubiquitin-like proteins with their targets, is able to protect stalled forks. To this purpose, we took advantage of the U2OS Flp-In T-REx system to develop cell lines expressing the siRNA-resistant forms of ISG15 wild type (WT) and ΔGG. Using this system, we can deplete endogenous ISG15 and appreciate the effect of the complementation with the exogenous constructs upon doxycycline induction. While the re-expression of ISG15 WT restores fork stability in ISG15-depleted cells, the ΔGG shows robust degradation of stalled forks (Fig. 1o, p). Taken together, these results strongly indicate that ISG15 conjugation is required to stabilize nascent DNA.

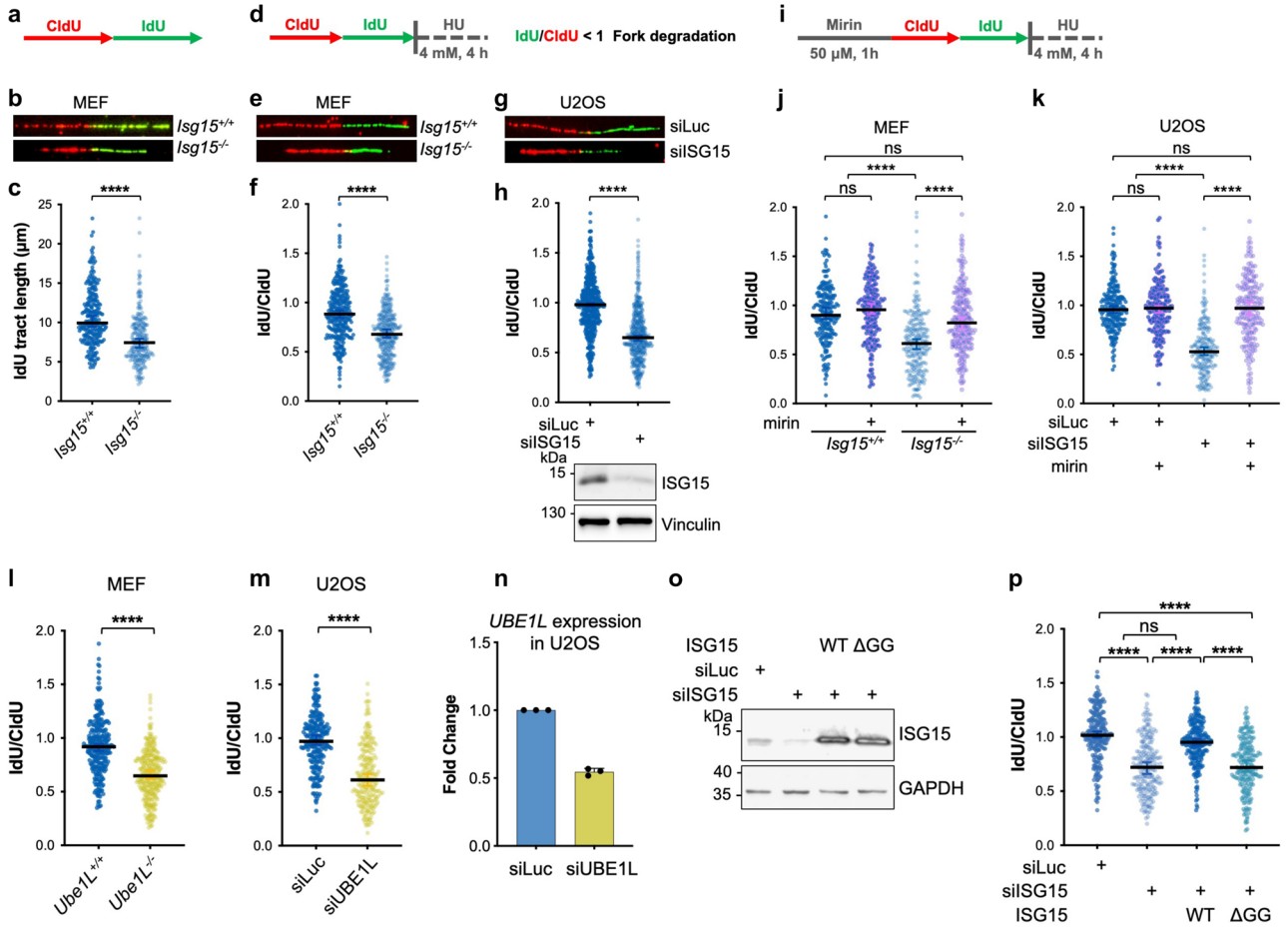

**Fig. 1 | Loss of ISG15 and UBE1L results in DNA replication fork instability.**
**a–c** Experimental workflow, representative images and size distribution of IdU tract lengths in *Isg15+/+* and *Isg15-/-* MEFs from three independent experiments (*Isg15+/+* n = 241 and *Isg15-/-* n = 244). **d** Schematic diagram of DNA replication fork degradation assay. **e–h** Representative images of fibers and IdU/CldU ratio analysis for the indicated conditions. In (**f**, **h**), fibers were analyzed in three and four independent experiments, respectively (*Isg15+/+* n = 288, *Isg15-/-* n = 315, siLuc n = 435 and siISG15 n = 440). Western blotting reveals the ISG15 expression. Vinculin immunoblot is used as loading control. **i–k** Experimental workflow and IdU/CldU ratio analysis from two independent experiments in MEFs (*Isg15+/+* n = 205, *Isg15+/+* +mirin n = 202, *Isg15-/-* n = 232, *Isg15-/-* +mirin n = 240), and U2OS cells (siLuc n = 203, siLuc +mirin n = 202, siISG15 n = 203 and siISG15 +mirin n = 203). **l, m** IdU/CldU ratio

analysis from three independent experiments in MEFs (*Ube1L+/+* n = 265, *Ube1L-/-* n = 304) and U2OS cells (siLuc n = 233, siUBE1L n = 228). **n** *UBE1L* mRNA expression in U2OS cells upon indicated siRNA treatment measured by qPCR (n = 3) corresponding to (**m**). Data are represented as mean + SD. **o** U2OS Flp-In T-REx cells expressing His-tagged ISG15, ISG15-ΔGG or the empty vector (EV), after doxycycline induction (1 μg/mL, 48 h) and optional treatment with siISG15. Immunoblot shows ISG15 protein levels and the loading control (GAPDH). **p** IdU/CldU ratio as in (**o**) from two independent experiments (EV siLuc n = 220, EV siISG15 n = 222, WT siISG15 n = 217 and ΔGG siISG15 n = 200). **c, f, l, m** Median value with 95% confidence interval (CI) is shown. Two-tailed Mann–Whitney test was performed; ****P < 0.0001. **j, k, p** Median value with 95% CI is shown. Two-tailed Kruskal–Wallis test was performed; ****P < 0.0001. Source data are provided as a Source data file.

## IFNβ fully restores the stability of stalled replication forks in BRCA1/2-deficient cells

The unexpected observation that basal levels of ISG15 and ISGylation are required for replication fork stability prompted us to test the effects of their upregulation in genetic contexts—such as the BRCA1/2-deficiencies—that are characterized by extensive degradation of newly synthesized DNA upon fork stalling. First, we tested the effect of low dose of IFNβ treatment—known to upregulate ISG15[13]—on the stability of nascent DNA strands. As previously reported[3], cells depleted of BRCA2 (siBRCA2) show excessive degradation of the nascent DNA compared to control cells (siLuc; Fig. 2a–c and Supplementary Fig. 2a). Remarkably, pre-treatment of cells with low dose of IFNβ (30 U/mL, 2 h) completely restores the stability of the forks (siBRCA2 + IFNβ). A similar effect was also observed in BRCA1-depleted cells (Fig. 2d, e), indicating a general effect of IFNβ treatment in promoting the stability of newly synthesized DNA in BRCA1/2-deficient cells. To extend our investigations into a more clinically relevant context, we tested the effect of IFNβ treatment on pancreatic adenocarcinoma cells (CAPAN-1) and triple-negative breast cancer cells (SUM149PT and MDA-MB-

436) carrying BRCA1 or BRCA2 defects and we invariably obtained the same result (Fig. 2f–k).

## ISG15 is required and sufficient for IFNβ-induced fork protection

To assess whether ISG15 is implicated in the effect of IFNβ treatment on fork integrity, we tested the effect of ISG15 depletion on IFNβ-mediated fork protection in BRCA2-deficient cells. Remarkably, reduction of ISG15 levels in U2OS cells completely reversed the effect of IFNβ on fork stability, clearly revealing the essential role of ISG15 in IFNβ-mediated stalled fork protection (Fig. 3a). Furthermore, we extended these analyses to MEFs and confirmed that loss of BRCA2 results in extensive degradation of stalled replication forks similarly to human cells. Notably, fork degradation in these MEFs is reversed by the treatment with mouse IFNβ and is completely dependent on ISG15 expression, indicating an evolutionary conserved function of type I IFN and ISG15 in the replication process (Fig. 3b and Supplementary Fig. 2b, c).

We next addressed whether the sole upregulation of ISG15 is sufficient to induce this phenotype, in absence of IFNβ stimulation. We used the ISG15 knockout (*ISG15-/-*) Flp-In TREx U2OS cells that we

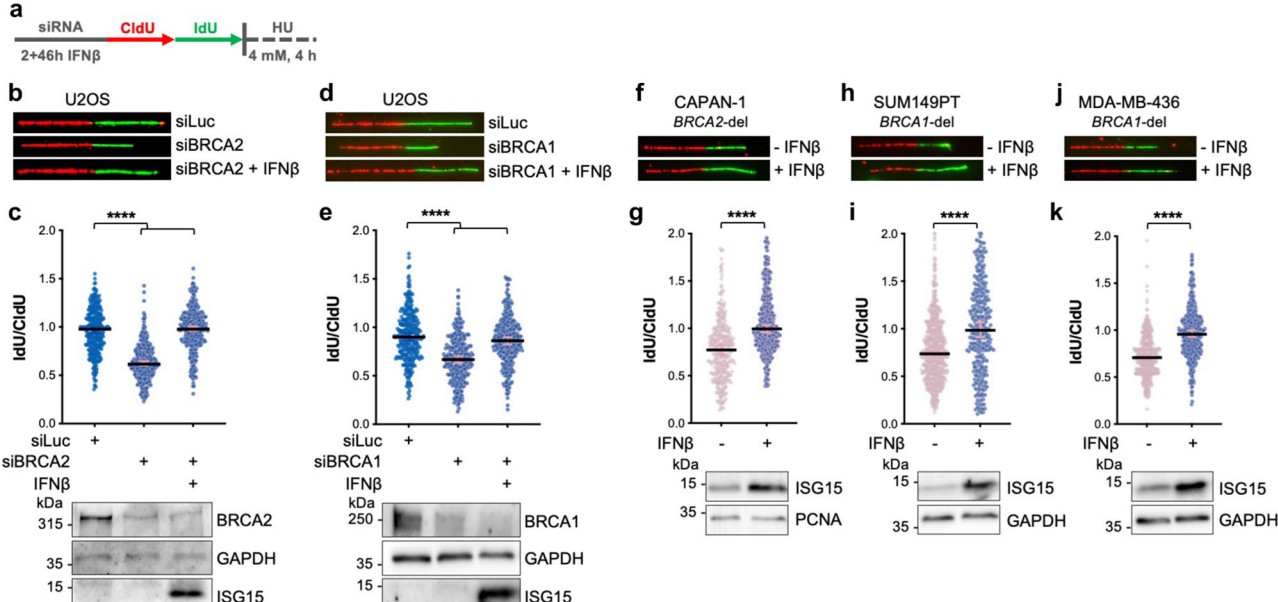

**Fig. 2 | IFNβ treatment promotes DNA replication fork protection in BRCA1/2 depleted cells. a** Schematic diagram of DNA replication fork degradation assay. **b, c** Representative images of fibers and IdU/CldU ratio analysis for the indicated conditions. Fibers were analyzed in three independent experiments (siLuc $n$ = 308, siBRCA2 $n$ = 307 and siBRCA2 +IFNβ $n$ = 311). Vinculin shows equal loading and ISG15 induction reveals activation of IFNβ pathway. **d, e** Representative images of fibers, IdU/CldU ratio analysis and BRCA1 and ISG15 protein expression for the indicated conditions. Fibers were analyzed in three independent experiments (siLuc $n$ = 303, siBRCA1 $n$ = 315 and siBRCA1 +IFNβ $n$ = 336). **f–k** Representative images of fibers and IdU/CldU ratio analysis in CAPAN-1, SUM149PT and MDA-MB-436 cells for the indicated conditions along with ISG15 protein expression. Fibers were analyzed in three independent experiments (CAPAN-1: -IFNβ $n$ = 335 and +IFNβ $n$ = 366; SUM149PT: -IFNβ $n$ = 629 and +IFNβ $n$ = 319; MDA-MB-436: -IFNβ $n$ = 447 and +IFNβ $n$ = 409) The lower panels show ISG15 expression along with the loading controls. **c, e** Median value with 95% confidence interval (CI) is shown. Two-tailed Kruskal–Wallis test was performed; ****$P$ < 0.0001. **g, i, k** Median value with 95% CI is shown. Two-tailed Mann–Whitney test was performed; ****$P$ < 0.0001. Source data are provided as a Source data file.

previously described[13], which re-express FLAG-ISG15 upon doxycycline induction, and found that the sole expression ISG15 can restore fork protection in BRCA2-deficient cells (Fig. 3c). Moreover, we engineered SUM149PT cells by lentiviral transduction to stably integrate a plasmid expressing MYC-ISG15 upon doxycycline treatment and confirmed that the expression of ISG15 is sufficient to restore fork stability in BRCA1-defective contexts (Fig. 3d).

## The ISG15 conjugation machinery is required for IFNβ-induced fork protection

To investigate whether the ISGylating enzymes (Fig. 3e) are also required for the restoration of fork stability in BRCA2-depleted cells, we first tested the effect of genetic ablation of UBE1L in MEFs ($Ube1L^{-/-}$) and found that it is essential for the IFNβ-dependent fork protection (Fig. 3f and Supplementary Fig. 2d, e). The two major ISG15 ligases promoting ISGylation are HERC5 and TRIM25[26,27]. Since HERC5 is mainly associated with innate antiviral response and has been reported to be rather promiscuous in terms of target specificity[28], we first focused on TRIM25 and found that loss of TRIM25 in MEFs ($Trim25^{-/-}$) prevented restoration of replication fork stability upon IFNβ stimulation (Fig. 3g and Supplementary Fig. 2f, g). Similar results were obtained in U2OS cells where IFNβ-dependent fork stabilization in BRCA2-depleted cells is reversed by loss of ISG15, UBE1L or TRIM25 (Supplementary Fig. 2h–j). Conversely, we observed no contribution of HERC5 in IFNβ-mediated fork protection in BRCA2-depleted U2OS cells (Supplementary Fig. 2h). Altogether these results show that ISG15 conjugation is required for IFNβ-mediated fork protection in BRCA2-deficient context.

## IFNβ does not rescue RAD51 *foci* in BRCA1/2-deficient cancer cells

BRCA1 and BRCA2 are important to promote homology-directed DNA repair, by promoting the loading of RAD51 onto chromatin, a key step

in HR. Consequently, RAD51 accumulation in discrete nuclear foci following DNA damage is largely impaired in BRCA1/2-deficient cells. To better assess the effect of the IFNβ/ISG15 system in these genetic contexts, we tested whether its upregulation exerts any effect on the restoration of chromatin loading of RAD51. Thus, we monitored the formation of RAD51 foci upon etoposide treatment (ETO) in BRCA1-defective cells (SUM149PT). We observed that the accumulation of RAD51 foci was highly impaired in ETO-treated cells and IFNβ treatment did not show any effect on the localization of RAD51 (Supplementary Fig. 3a–c). We obtained analogous results using the BRCA2-deficient ($Brca2^{-/-}$) mouse mammary tumor cell line KB2P (clone 1.21[29]), confirming that treatment with IFNβ did not restore RAD51 foci (Supplementary Fig. 3d–f). Next, we performed similar experiments in U2OS cells upon transient depletion of BRCA1 or BRCA2. RAD51 foci induced by ETO treatment and by ionizing radiation (IR) were readily detectable in control cells (siLuc) and, expectedly, highly reduced both in BRCA1- and BRCA2-depleted cells. Here, IFNβ treatment partially restored RAD51 foci as compared to BRCA1/2-proficient cells (Supplementary Fig. 3g–l), probably reflecting the residual amount of BRCA1 and BRCA2 still present in the cells. Overall, these results suggest that IFNβ treatment could not significantly restore the accumulation of RAD51 foci induced by DNA damage in BRCA1/2-deficient cells.

## IFNβ restores the viability of BRCA2-deficient mESCs via ISG15 conjugation

*BRCA1* and *BRCA2* genes are essential in mammals, and their knockout leads to early embryonic lethality in mice[30–32]. It has been reported that the survival of *Brca2*-deficient mESCs can be promoted by restoration of fork protection, while HR is still impaired[33]. Hence, we asked whether IFNβ/ISG15, by promoting replication fork protection in BRCA2-deficient cells, can also restore viability of mESCs. To address this, we used PL2F7 mESCs that carry one functionally null and one conditional

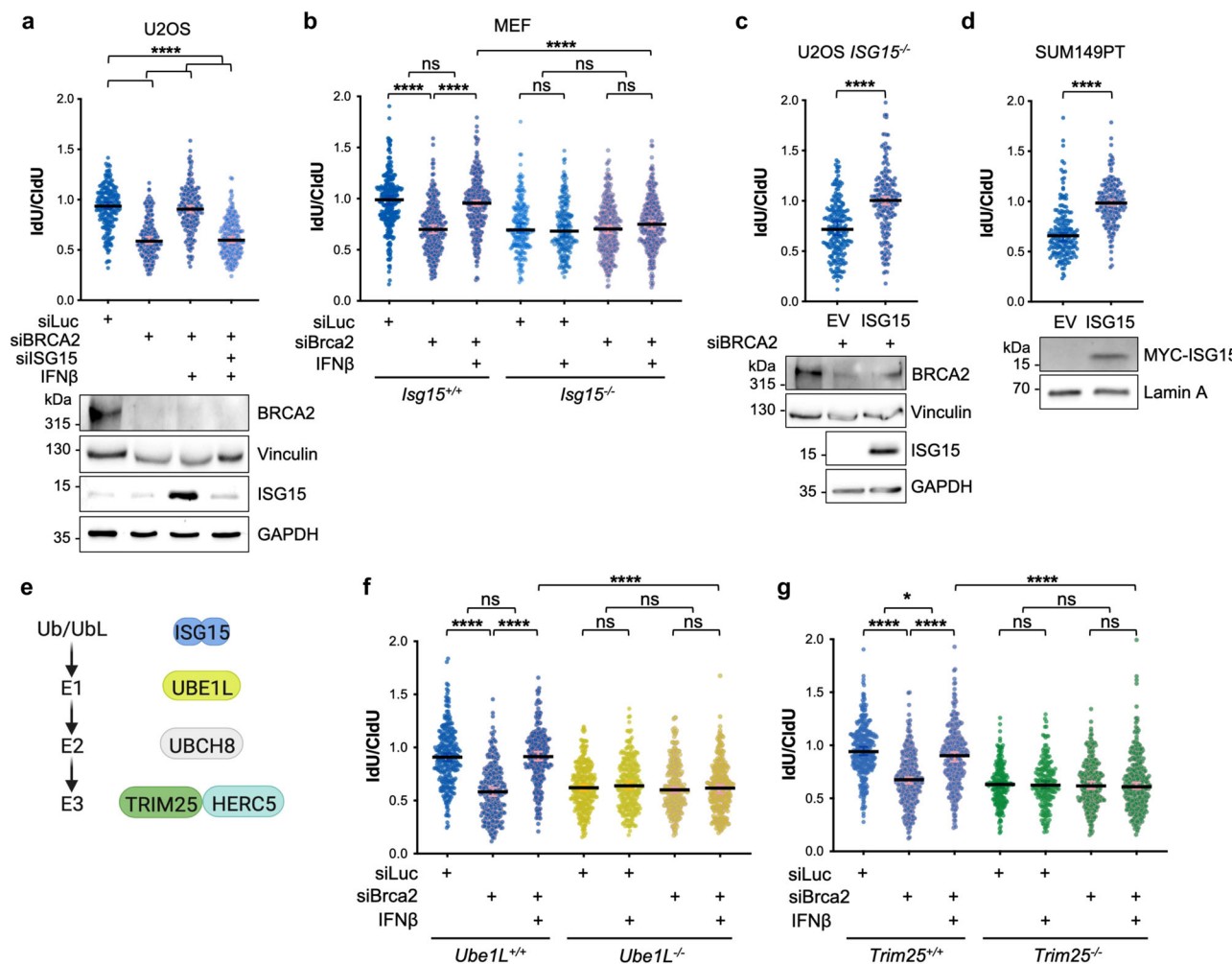

**Fig. 3 | IFNβ and ISG15 restore fork protection in BRCA-deficient cells via ISG15 conjugation. a** IdU/CldU ratio analysis for the indicated conditions along with ISG15 and BRCA2 expression and the loading controls. Fibers were analyzed in two independent experiments (siLuc *n* = 209, siBRCA2 *n* = 203, siBRCA2 +IFNβ *n* = 205 and siISG15 + siBRCA2 +IFNβ *n* = 218). **b** IdU/CldU ratio analysis for the indicated conditions. Fibers were analyzed in three independent experiments (*Isg15*⁺ᐟ⁺ siLuc *n* = 287, *Isg15*⁺ᐟ⁺ siLuc +IFNβ *n* = 201, *Isg15*⁺ᐟ⁺ siBrca2 *n* = 292, *Isg15*⁺ᐟ⁺ siBrca2 +IFNβ *n* = 330, *Isg15*⁻ᐟ⁻ siLuc *n* = 204, *Isg15*⁻ᐟ⁻ siLuc +IFNβ *n* = 204, *Isg15*⁻ᐟ⁻ siBrca2 *n* = 313 and *Isg15*⁻ᐟ⁻ siBrca2 +IFNβ *n* = 315). **c** IdU/CldU ratio analysis in in BRCA2 depleted *ISG15*⁻ᐟ⁻ U2OS Flp-In T-REx cells expressing empty vector (EV) or FLAG-ISG15 along with BRCA2 and ISG15 protein expression and the loading controls. Fibers were analyzed in two independent experiments (EV siBRCA2 *n* = 200 and ISG15 siBRCA2 *n* = 221). **d** IdU/CldU ratio analysis in SUM149PT cells expressing empty vector (EV) or MYC-ISG15 (*n* = 3) along with MYC-ISG15 protein expression and the loading control. Fibers were analyzed in three independent experiments (EV *n* = 314 and ISG15 *n* = 372). **e** Schematic of the enzymes involved in ISG15 conjugation. **f**, **g** IdU/CldU ratio analysis for the indicated conditions. Fibers were analyzed in three independent experiments (*Ube1L*⁺ᐟ⁺ siLuc *n* = 271, *Ube1L*⁺ᐟ⁺ siBrca2 *n* = 277, *Ube1L*⁺ᐟ⁺ siBrca2 +IFNβ *n* = 302, *Ube1L*⁻ᐟ⁻ siLuc *n* = 289, *Ube1L*⁻ᐟ⁻ siLuc +IFNβ *n* = 277, *Ube1L*⁻ᐟ⁻ siBrca2 *n* = 276 and *Ube1L*⁻ᐟ⁻ siBrca2 +IFNβ *n* = 275; *Trim25*⁺ᐟ⁺ siLuc *n* = 303, *Trim25*⁺ᐟ⁺ siBrca2 *n* = 303, *Trim25*⁺ᐟ⁺ siBrca2 +IFNβ *n* = 338, *Trim25*⁻ᐟ⁻ siLuc *n* = 203, *Trim25*⁻ᐟ⁻ siLuc +IFNβ *n* = 201, *Trim25*⁻ᐟ⁻ siBrca2 *n* = 290 and *Trim25*⁻ᐟ⁻ siBrca2 +IFNβ *n* = 301). **a**, **b**, **f**, **g** Median value with 95% CI is shown. Two-tailed Kruskal−Wallis test was performed; *P = 0.0267 ****P < 0.0001. **c**, **d** Median value with 95% CI is shown. Two-tailed Mann−Whitney test was performed; ****P < 0.0001. Source data are provided as a Source data file.

allele of *Brca2* (*Brca2^flox/-*; Fig. 4a and Supplementary Fig. 4a). Here, the transfection with CRE recombinase generates a functional *HPRT* minigene, which in principle allows these cells to grow in HAT (hypoxanthine, aminopterin and thymidine) medium, and promotes the deletion of the conditional allele, leading to complete loss of BRCA2 and thereby to lethality[34,35]. To test the effect of IFNβ on the viability of mESCs, we treated these cells with mouse IFNβ, either in continuous for 48 h or for 2 h pulse followed by 46 h chase, prior to CRE transfection and selection in HAT medium (Fig. 4b). Genotyping of the very few surviving colonies did not reveal any *Brca2^-/-* clones in untreated cells (DMSO), confirming the essential role of BRCA2 in mESCs viability. In contrast, pre-treatment with IFNβ led to a remarkable number of viable *Brca2^-/-* clones, exceeding 30%, indicating that the upregulation of IFNβ signaling efficiently rescues the viability of *Brca2*-deficient mESCs (Fig. 4c, d). Consistently, also in this system we observed rescue of replication fork protection following IFNβ,

although to a lesser extent compared to other systems (Supplementary Fig. 4b, c). To investigate the possible role of the ISG15 system in this context, we monitored the effect of loss of ISG15, or of the ISGylating enzymes UBE1L and TRIM25, on the IFNβ-induced viability of *Brca2^-/-* cells. Remarkably, even upon IFNβ treatment, we obtained no clones that are *Brca2^-/-* in cells depleted of ISG15, UBE1L or TRIM25 (Fig. 4e, f and Supplementary Fig. 4d, e), highlighting the essential role of these genes−and of ISGylation in general−in the IFNβ-mediated rescue of viability upon BRCA2 loss.

**TOP1 is required to promote IFNβ-mediated fork protection in BRCA1/2-defective cells**

ISG15 was the first ubiquitin-like protein identified[36] and several hundreds of its potential targets have been reported in the last two decades. However, for very few of them ISGylation has been validated or the exact modification site identified, making it difficult to evaluate the

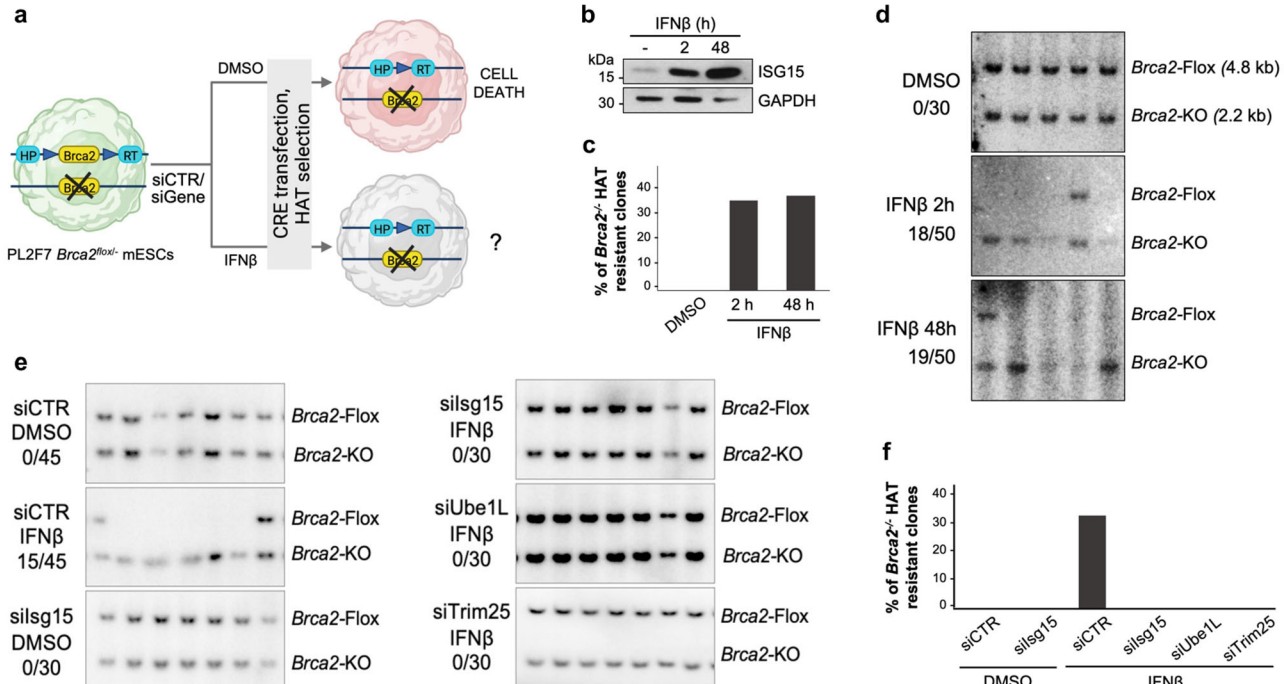

**Fig. 4 | Upregulation of IFNβ/ISG15 restores viability in BRCA2-deficient mESCs.** **a** Schematic for testing conditions for generation of *Brca2⁻ᐟ⁻* in PL2F7 (created with BioRender.com). **b** ISG15 protein expression in PL2F7 cells after optional treatment with IFNβ (30 U/mL) either for 48 h in continuous or for 2 h pulse and 46 h chase. **c, d** Percentage of *Brca2⁻ᐟ⁻* HAT resistant clones and representative Southern blot showing *Brca2ᶠˡᵒˣᐟ⁻* or *Brca2⁻ᐟ⁻* mESC upon ± IFNβ pre-treatment as in (**b**). **e, f** Ratio of number of rescued clones and total numbers of HAT resistant clones analyzed, and representative Southern blot showing *Brca2ᶠˡᵒˣᐟ⁻* or *Brca2⁻ᐟ⁻* mESC upon ± IFNβ (30 U/mL, 2 h) pre-treatment along with depletion of ISG15, UBE1L or TRIM25 by siRNA treatment. Source data are provided as a Source data file.

molecular impact of ISGylation[37]. Here, to shed light on the mechanism underpinning the effect of IFNβ on the stability of replication forks, we assessed how chromatin composition varies under conditions in which ISG15 promotes fork stability. To avoid IFNβ treatment that might induce perturbations in protein expression and chromatin composition that are unrelated to the effect on DNA replication, we used engineered SUM149PT cell lines expressing MYC-ISG15 or the empty vector (EV) in a doxycycline-inducible manner (Fig. 5a, b). We isolated chromatin fractions in both cell lines, following optional treatment with HU (4 mM, 4 h); the experiment was performed in biological triplicates for statistical significance. Mass spectrometry analysis of factors associated with chromatin fractions identified more than 2000 proteins, of which 191 appeared differently regulated upon ISG15 induction (Supplementary Data 1). Proteins with a fold change of less than −/+0.5 were not considered further. The remaining factors—72 upregulated and 62 downregulated—were then cross-referenced with the list of putative ISG15 substrates identified in several mass spectrometry studies reported in the literature[20,23,35–47]. Interestingly, among them the topoisomerase-1 (TOP1; Fig. 5c) caught our attention, since it was identified as potential ISG15 target in three different reports[20,38,44] and for its crucial role in replication dynamics and genome integrity[48,49].

As the low concentration of IFNβ (30 U/mL) used in our experiments does not lead to detectable formation of ISGylated conjugates (Supplementary Fig. 5a, b), we tested whether TOP1 can be targeted by ISGylation by ectopically expressing the ISGylating enzymes UBE1L, UBCH8 and TRIM25 in HEK293 cells, together with FLAG-ISG15 WT and the conjugation-defective mutant ΔGG. FLAG immunoprecipitation performed under stringent conditions followed by Western blot revealed the presence of TOP1 in the sample where ISG15 WT was present but not in the ΔGG mutant (Fig. 5d). To further validate this result, we performed the reciprocal experiment, i.e., we immunopurified FLAG-TOP1 from cells expressing the ISGylation machinery as in Fig. 5d and monitored its ISGylation status by using ISG15 antibody. In the presence of ISG15 WT, we could observe a signal corresponding

to ISGylated TOP1, which is missing in cells expressing the conjugation-defective form (Fig. 5e), suggesting that TOP1 can undergo ISG15 conjugation under these conditions.

Next, we investigated the possible contribution of TOP1 to the fork stabilization promoted by IFNβ in BRCA1/2-deficient cells. We first performed a DNA fiber assay in SUM149PT cells to measure the degradation of HU-induced stalled replication forks upon optional treatment with IFNβ and TOP1 depletion. As expected, these cells show extensive degradation of the newly synthesized DNA, which is reversed by IFNβ treatment. Remarkably, TOP1 depletion, albeit partial, abrogates this effect, resulting in degradation of the forks to levels comparable to those of siLuc-treated cells (Fig. 5f). Similarly, TOP1 is essential to promote stalled forks stability induced by IFNβ in U2OS cells upon BRCA2 depletion (Fig. 5g).

We have previously reported that IFNβ/ISG15 signaling promotes high speed of DNA replication fork progression[13]. In such a scenario, the role of (ISGylated) TOP1 might be to help resolving the torsional stress ahead of the forks, assisting the accelerated fork restart linked to high ISG15 levels and diminishing the half-life of the regressed arms, which are established entry points for fork degradation in BRCA1/2-deficient cells[23,24,50,51]. To test this hypothesis and assess whether this function of TOP1 can be extended to other contexts, we investigated another independent condition, i.e., PARP inhibition, which suppresses fork degradation in BRCA2-deficient cells by accelerating restart of reversed forks[24,52,53]. Notably, we found that TOP1 depletion completely reverts this effect, indicating a general effect of TOP1 to rescue fork integrity in BRCA2-defective cells by altering dynamics and architecture of replication intermediates (Fig. 5h).

## ISG15 is upregulated in BRCA1/2-deficient cells and is required for their fitness

Several lines of evidence indicate that conditions of replication fork instability and DNA damage lead to the accumulation of DNA fragments in the cytosol, which activate type I IFN via the cGAS/STING

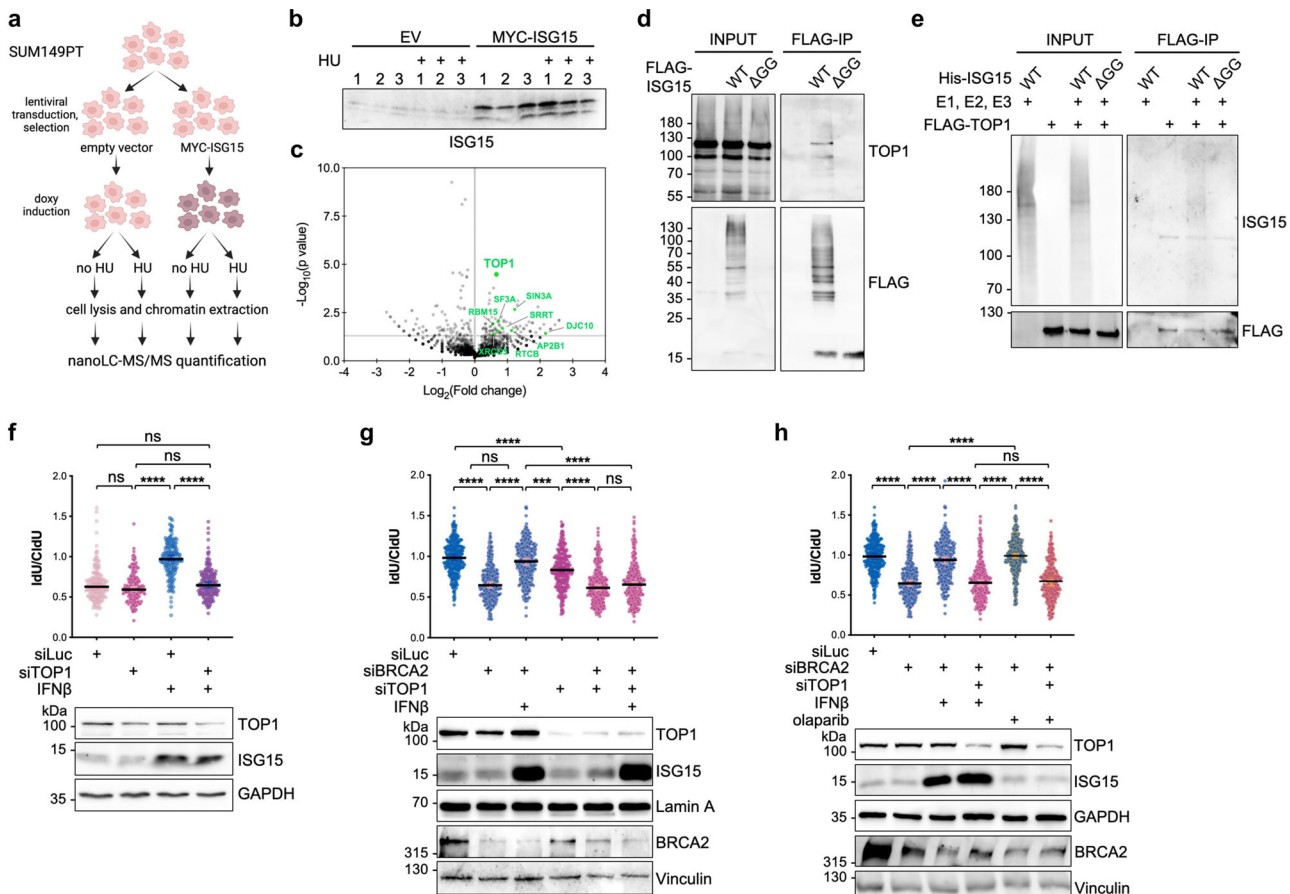

**Fig. 5 | TOP1 is required for IFNβ-mediated restoration of replication fork stability in BRCA-defective cells. a, b** Schematic workflow of the mass spectrometry analysis (created with BioRender.com) and immunoblot showing ISG15 protein levels. Three biological replicates were analyzed for each condition. **c** Volcano plot revealing the distribution of mass spectrometry hits from EV and MYC-ISG15 expressing SUM149PT cells treated with HU. The horizontal axis shows Log$_2$ fold change of protein abundance (EV vs ISG15). The vertical axis shows −Log$_{10}$ of the Fisher's exact test $P$ value. Points with a Log$_2$(Fold change) >0 indicate proteins that are enriched in cells expressing MYC-ISG15. The horizontal gray line represents the p-value threshold at 0.05. Points below this line ($P > 0.05$) indicate proteins that do not meet the statistical significance criteria. Putative targets of ISGylation that appear at least in two different studies are highlighted in green. **d** Western blot of total cell extracts (input) and FLAG immunoprecipitates from HEK293T cells transfected with FLAG-tagged ISG15 WT and ΔGG, together with the ISGylation machinery (E1, E2, E3), UBE1L, UBCH8 and TRIM25. **e** The reciprocal

experiment is performed by transfecting HEK293T cells with FLAG-tagged TOP1, His-tagged ISG15 and ΔGG, together with the ISGylation machinery (E1, E2, E3), and proceeding with FLAG-immunoprecipitation as in (**d**). **f, g** IdU/CldU ratio analysis in SUM149PT (f) and U2OS (g) cells for the indicated conditions along with TOP1, ISG15 and BRCA2 protein expression. In (**f, g**), fibers were analyzed in two and three independent experiments, respectively (SUM149PT: siLuc $n = 200$, siTOP1 $n = 200$, siLuc +IFNβ $n = 203$ and siTOP1 +IFNβ $n = 203$; U2OS: siLuc $n = 318$, siBRCA2 $n = 322$, siBRCA2 +IFNβ $n = 326$, siTOP1 $n = 304$, siTOP1 + siBRCA2 $n = 284$ and siTOP1 + siBRCA2 +IFNβ $n = 303$). **h** IdU/CldU ratio analysis of three independent experiments for the indicated conditions (siLuc $n = 318$, siBRCA2 $n = 322$, siBRCA2 +IFNβ $n = 326$, siTOP1 + siBRCA2 +IFNβ $n = 303$, siBRCA2 +olaparib $n = 300$ and siTOP1 + siBRCA2 +olaparib $n = 298$) along with TOP1, ISG15 and BRCA2 protein expression. **f–h** Median value with 95% CI is shown. Two-tailed Kruskal–Wallis test was performed; ***$P = 0.0001$; ****$P < 0.0001$. Source data are provided as a Source data file.

pathway[54]. As IFNβ strongly induces ISG15 expression, we reasoned that in genetic backgrounds characterized by genomic instability due to replication stress and DNA repair defects (such as BRCA1/2 deficiencies), ISG15 expression might be elevated. To test this prediction, we compared ISG15 protein levels in isogenic pairs of BRCA1/2-deficient and -proficient cancer cells and found a consistent upregulation of ISG15 in cells carrying defects in either BRCA1 or BRCA2 (Fig. 6a), suggesting that it may be required for the fitness of BRCA1/2-deficient cells. Indeed, ISG15 depletion markedly reduces the viability of MDA-MB-436 cells, which carry mutations in *the BRCA1* gene, while MDA-MB-436 cells reconstituted with wild type *BRCA1*[55] show only limited growth defects (Fig. 6b, c and Supplementary Fig. 6a).

### Upregulation of ISG15 confers chemo-resistance to BRCA2-deficient cancer cells

Alterations in *BRCA1/2* genes are associated with marked genomic instability and cancer predisposition. On the other hand, these

conditions provide a window of opportunity for therapeutic intervention, since these tumor cells are exquisitely sensitive to chemotherapeutic drugs, such as cisplatin and PARP1 inhibitors[56–58]. Unfortunately, BRCA1/2-deficient cancer cells very often acquire drug resistance, by means of different mechanisms, including reversion of *BRCA1/2* mutations, restoration of HR and of replication fork stability[59]. As we observed that upregulation of ISG15 results in restoration of fork protection in BRCA1/2-deficient contexts, we asked whether it could also reduce sensitivity of these cells to cisplatin. To this purpose, we took advantage of the BRCA2-deficient KB2P cell line and its isogenic counterpart where the cell line has been reconstituted with human BRCA2. These cells were further engineered to obtain doxycycline-inducible expression of MYC-tagged mouse ISG15 (Fig. 6d). In line with our previous results, IFNβ-mediated ISG15 induction restores protection of stalled forks in KB2P cells, although only partially (Supplementary Fig. 6b). Remarkably, cells defective in BRCA2 (*Brca2$^{-/-}$*) show high sensitivity to cisplatin as compared to BRCA2-proficient cells, but

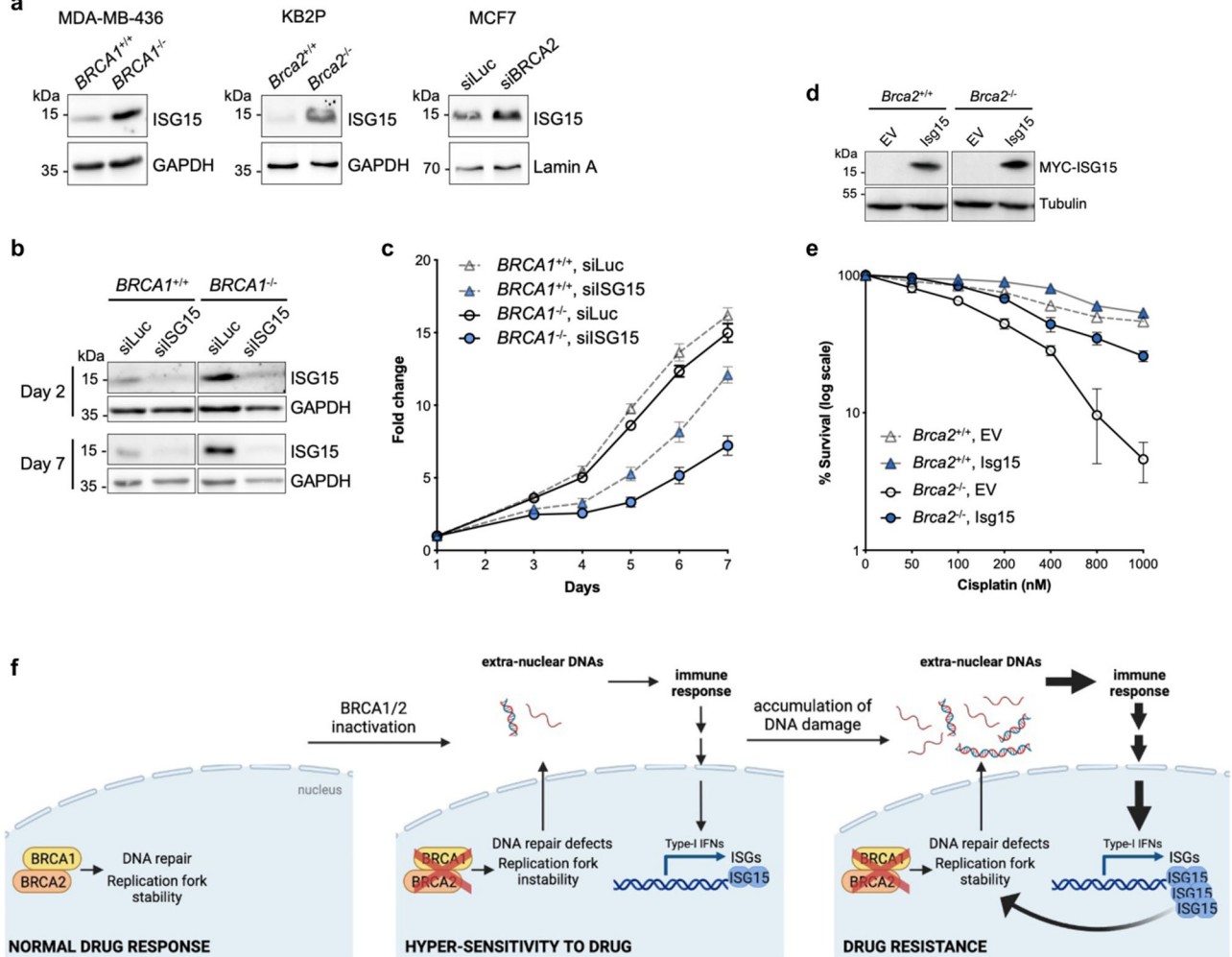

**Fig. 6 | ISG15 is upregulated in BRCA1/2-deficient cells and is required for their fitness and reduced drug sensitivity. a** Immunoblot showing ISG15 protein levels in MDA-MB-436, KB2P and MCF7 cells upon BRCA1 or BRCA2 deletion or depletion. **b** Immunoblot showing ISG15 protein levels in BRCA1-proficient or -deficient MDA-MB-436 cells at day 2 and day 7 post siISG15 treatment. GAPDH was used as loading control. **c** Cell proliferation graph showing fold change of viable cells (mean with SEM) at indicated days normalized to cells at day 1 after seeding (*n* = 3). **d** Immunoblot showing ISG15 protein levels in KB2P cells corresponding to Fig. 6e. Tubulin, loading control. **e** Graph showing the percentage of surviving clones upon treatment with cisplatin with indicated doses in BRCA2-proficient (*Brca2*+/+ *p53*-/-) and -deficient (*Brca2*-/- *p53*-/-) KB2P mouse cells upon ISG15 overexpression (*n* = 3; error bars represent mean ± SEM). **f** Model depicting the contribution of IFNβ/ISG15

to drug response (created with BioRender.com). Normal cells, characterized by efficient DNA repair and replication fork stability, show no sensitivity to chemotherapeutic drugs. Following the inactivation of DNA repair/replication factors, such as BRCA1/2, DNA repair and replication processes are impaired, resulting in genomic instability and the release of nucleic acids species into the cytosol, which induces mild activation of the immune response. Over time, the accumulation of extra-nuclear DNAs leads to massive activation of the immune response, ultimately fostering a strong induction of the IFN-stimulated genes (ISGs), including ISG15 and its conjugation system, which restores stability of replication fork—via TOP1—and favors the acquisition of drug resistance. Source data are provided as a Source data file.

upregulation of ISG15 dramatically reduces cisplatin sensitivity (Fig. 6e and Supplementary Fig. 6c).

## Discussion

It is well established that replication fork instability leads to accumulation of cytosolic nucleic acids, mimicking pathogen infection and therefore triggering the IFN-mediated immune response and inflammation. What is still unclear is how this inflammatory response in turn affects different aspects of DNA metabolism. Our work aims to close the circle and investigate how this IFN-mediated inflammatory response regulates DNA replication and repair processes, possibly impacting on cancer cell fitness and therapy response.

Tumors deficient in BRCA1/2 are associated with marked genomic instability, due to their pivotal functions in DNA repair via HR and DNA replication fork stability, which make them extremely sensitive to chemotherapeutic agents. Unfortunately, most patients develop drug

resistance through different mechanisms, including restoration of HR and replication fork protection[33,60]. Here we show that activation of the IFN pathway restores the stability of nascent DNA in BRCA1/2-deficient contexts. Despite pleiotropic effects and numerous targets of IFN signaling, this effect is completely dependent on a single factor of the IFN signaling—i.e., the ISG15 system—indicating a key role for protein ISGylation in replication fork dynamics. ISG15-mediated fork protection is beneficial to cancer cells; indeed, the sole upregulation of ISG15 increases resistance of BRCA2-deficient mouse breast cancer cells to platinum-derived chemotherapeutic agents. These findings offer an explanation for the controversial data on the effect of type I IFN in cancer therapy. It is well established that the efficacy of several therapeutic strategies against cancer, including cytotoxic drugs, radiotherapy and targeted immunotherapies, depends on type I IFN signaling[61]. On the other hand, several studies indicate that IFN signaling may induce resistance to DNA damage and radiotherapy[62] and

identified an IFN-related DNA damage resistance signature (IRDS)—including IFN response genes and ISG15 itself—correlating with acquired chemo- and radio-resistance[63,64]. Our data show that ISG15 can promote drug resistance in specific genetic contexts, i.e., BRCAness, through the stabilization of nascent DNA strands. Intriguingly, while ISG15-mediated restoration of fork protection has been consistently observed in all cellular systems we have tested, the acquisition of drug resistance appears to be rather tumor and cell type-specific, probably depending on the complexity of the genetic background. This variability is likely to reflect differential expression of key factors, which are positively or negatively regulated by ISG15 and ISGylating enzymes. Future investigations are needed to identify those factors that, in combination with the upregulation of ISG15, are required to alter fitness and drug sensitivity of BRCA1/2-mutated cancer cells.

We have previously shown that ISG15 upregulation accelerates DNA replication fork progression by promoting the activity of the RECQ1 helicase to restart replication forks that spontaneously stall in unperturbed cells[13]. We hypothesize that an ISG15-mediated increase in replication rate counteracts replication fork reversal, a protective mechanism that remodels replication forks into four-way-junctions in response to replication stress[65], in favor of RECQ1-mediated fork restart. As fork degradation in BRCA-defective cells requires reversed forks as entry points for deregulated nucleolytic activities[23,24,50,51], we propose that ISG15-mediated fork restart rescues fork integrity in BRCA-defective cells by counteracting reversed fork accumulation. In an effort to provide more mechanistic understanding into this regulation, we also found that ISG15-mediated restoration of fork integrity in BRCA-defective cells strictly requires TOP1. This genetic requirement may reflect the need to resolve topological constraints, in order for ISG15 to foster fork restart, thereby counteracting fork reversal and degradation. This interpretation is in line with original findings that had identified fork reversal as a mechanism to redistribute topological stress at stalled forks[66], implying the need to overcome these constraints when reversed forks are being restarted. Our data also implicate that the requirement for TOP1 activity may represent a promising and druggable vulnerability of BRCA1/2-defective tumors that have acquired chemoresistance via activated IFN signaling. This is also consistent with previous evidence from our lab and others that ISG15 upregulation sensitizes cells to TOP1 inhibitors[13,67].

Further emphasizing the functional relevance of IFN signaling upon BRCA defects, our experiments in mESC revealed that activation of the IFNβ/ISG15 system restores viability of BRCA2-deficient cells. Importantly, also in this system we observed that IFNβ treatment protects stalled forks from degradation but does not rescue RAD51 foci formation. This result suggests that, even in the absence of HR, the restoration of replication fork protection is sufficient to support the viability of BRCA2-defective mESCs. Similarly, previous studies showed that suppression of fork degradation and reversed fork protection (as upon depletion of PTIP) promote ESC viability without impacting on RAD51 foci[24,33,34]. In line with those observations, our data support the concept that defects in fork metabolism contribute to mESC lethality upon BRCA2 loss.

Notably, we observed that deficiency in *BRCA* genes increases *ISG15* expression, as compared to their syngeneic BRCA1/2-proficient counterparts, likely as a consequence of accumulated cytosolic DNA fragments and activation of the immune response. This consistent ISG15 upregulation is beneficial for BRCA-deficient cells, as its depletion dramatically reduces cell viability. Over time, sustained ISG15 expression—fostered by BRCA deficiency—may per se lead to restoration of fork protection and promote drug resistance, suggesting the intriguing hypothesis that BRCAness—as well as other genetic conditions linked to fork instability—are intrinsically prone to develop chemoresistance (Fig. 6f). It will be important to extend these intriguing observations to other HR deficiencies.

Taken together, our results identify the IFNβ/ISG15 pathway as a key modulator of DNA replication fork protection and help explain the complex role of type I IFN—and ISG15 itself—in cancer development and drug response. Moreover, our data implicate that the IFNβ/ISG15 system should be carefully considered as a target to potentiate classical chemotherapy or as a critical modulator for immunotherapeutic options, especially when combined with DNA replication interference.

## Methods

### Cell lines and cell culture
MEFs, U2OS (ATCC HTB-96), HEK293T (ATCC CRL-11268) and MCF7 (ATCC HTB-22) cells were cultured in DMEM supplemented with 10% FBS and 0.05 U penicillin/streptomycin. U2OS Flp-In TREx cells were cultured in DMEM supplemented with 10% FBS, 0.05 U penicillin/streptomycin, 10 μg/mL blasticidin and 100 μg/mL hygromycin B. SUM149PT cells (CVCL_3422) were cultured in adDMEM/F12 with 5% FBS, 0.05 U penicillin/streptomycin, 5 μg/mL of insulin (Sigma-Aldrich; I9278), 10 mM HEPES (Sigma-Aldrich; H0887), 2 mM L-glutamine (Gibco; 25030-024), 1 μg/mL of hydrocortisone (Sigma-Aldrich; H0888-1G). MDA-MB-436 *BRCA1[-/-]* and *BRCA1[+/+]* (reconstituted) cells were kindly gifted by Neil Johnson. Cells were maintained in RPMI media supplemented with 10% FBS, 0.05 U penicillin/streptomycin. CAPAN-1 cells (ATCC HTB-79) were cultured in DMEM supplemented with 20% FBS, 0.05 U penicillin/streptomycin. *Brca2[+/+]* and *Brca2[-/-]* mouse mammary tumor cells (KB2P 1.21) have been previously described[68]. Both *Brca2[+/+]* and *Brca2[-/-]* cell lines were cultured under low oxygen conditions (3% $O_2$, 5% $CO_2$, 37 °C) using DMEM supplemented with 10% FBS, 50 U/mL penicillin, 50 μg/mL streptomycin, 5 μg/mL insulin (Sigma-Aldrich; I0516), 5 ng/mL murine epidermal growth factor (Sigma-Aldrich; E4127), and 5 ng/mL cholera toxin (List biological laboratories; 9100B).

All mouse embryonic stem cells (mESCs) were cultured on top of mitotically inactive STO-neomycin-LIF-puromycin (SNLP) feeder cells in M15 media [knockout DMEM media (Life Technologies) supplemented with 15% FBS (GE Life Sciences-Hyclone), 0.00072% β-mercaptoethanol, penicillin (100 U/mL), streptomycin (100 μg/mL), and L-glutamine (0.292 mg/mL)] at 37 °C and 5% $CO_2$. PL2F7 cells expressing *BRCA2 R2336H* variant were generated by electroporating respective bacterial artificial chromosomes in PL2F7 cells[35,69]. PL2F7-*Brca2[-/-]*;*BRCA2(R2336H)* cells only express the hypomorphic allele. *Trim25[+/+]* and *Trim25[-/-]* MEFs were kindly gifted by Satoshi Inoue (Tokyo Metropolitan Institute of Gerontology, University of Tokyo, Japan). *Ube1L[+/+]* and *Ube1L[-/-]* MEFs were kindly gifted by Dong-Er Zhang (Moores Cancer Center, University of California, San Diego, USA). *Isg15[+/+]* and *Isg15[-/-]* MEFs were kindly gifted by Klaus-Peter Knobeloch (Institute of Neuropathology, University Clinic Freiburg, Germany).

### Antibodies
The list of antibodies used in this study is provided in the Supplementary Information as Supplementary Table 1.

### Lentiviral transduction
For lentivirus production, HEK293T cells were transfected with the packaging plasmids pVSV, pMDL, pREV and the expression plasmid pCW57.1-TRE-MYC-ISG15-IRES-EGFP or pCW57.1-TRE-IRES-EGFP (derived from pCW57.1-TRE kindly provided by Arnab Ray Chaudhuri) using jetPRIME® (Polyplus Transfection) according to manufacturer's instructions. The following day, medium was replaced with fresh DMEM medium. Lentivirus was collected 72 h after transfection and stored at −80 °C. Lentiviral titer was determined using FACS analysis (BD LSR II Fortessa) for GFP positive cells 72 h after transduction and doxycycline treatment (1 μg/mL). For stable cell line generation, SUM149PT cells grown on 6-well plates were transduced overnight with lentiviral particles (MOI = 1) in adDMEM/F12 supplemented with

10 µg/mL polybrene. Cells were washed 18 h after transduction and selected using blasticidin antibiotics.

## DNA fiber assay

Following the depletion of proteins of interest or IFNβ treatment, cells were sequentially pulse-labeled with 33 µM CldU (Sigma-Aldrich; C6891) and 339 µM IdU (Sigma-Aldrich; I7125) for 20 min, or in case of U2OS cells 30 min. Following, the cells were optionally treated with 4 mM HU (Sigma-Aldrich; H8627) for 4 h at 37 °C. Cells were collected and resuspended in cold PBS to a concentration of $2.5 \times 10^5$ labeled and $3.5 \times 10^5$ unlabeled cells per mL. Labeled cells were mixed 1:2 (v/v) with unlabeled cells and 4 µL cells were lysed for 9 min with 7.5 µL lysis buffer (200 nM Tris-HCl pH 7.4, 50 mM EDTA, 0.5% (w/v) SDS) directly on a glass slide. Slides were tilted at 30–45° to stretch the DNA fibers, air-dried, and fixed in 3:1 methanol/acetic acid overnight at 4 °C. The fibers were denatured with 2.5 M HCl for 1.5 h, washed with PBS and blocked for 40 min with 2% BSA/PBS-Tween. The CldU and IdU tracts were stained for 2.5 h with anti-BrdU primary antibodies recognizing CldU (1:500; Abcam; ab6326) and IdU (1:100; BD Biosciences; 347580), and for 1 h with secondary antibodies anti-mouse Alexa Fluor 488 (Invitrogen; A11001) and anti-rat Cy3 (Jackson Immuno Research; JAC712-166-153) in the dark. Coverslips were mounted using ProLong Gold Antifade Mountant. Images were acquired on a Leica DM6 B microscope at a lens-magnification of 63x and analyzed using ImageJ software (NIH). The pixel values were converted to µm (1 pixel corresponds to 0.146 µm) and IdU/CldU ratios were calculated as a measure of stalled replication fork degradation. Per sample, at least 200 individual fibers were scored. Data were pooled from independent experiments. The number of biological replicates is indicated in the figure legend. Statistical differences in IdU tract lengths or IdU/CldU ratios were determined by Kruskal–Wallis or Mann–Whitney test (GraphPad Prism 9).

## siRNA transfection

Cells were plated and transfected the following day for 48 or 72 h (as indicated below) using Oligofectamine transfection reagent (Invitrogen) according to the manufacturer's protocol.

Human cell lines were transfected with the following siRNAs:
siLuc (5′-CGUACGCGGAAUACUUCGAUUdTdT-3′);
siISG15 (50 nM, 72 h; 5′-GCAACGAAUUCCAGGUGUCdTdT-3′);
siUBE1L (50 nM, 72 h; 5′-UAGUGCUGGCGUCUCAGCUUCUCCUdTdT-3′);
siBRCA1 (40 nM, 48 h; 5′-GGAACCUGUCUCCACAAAGTT-3′);
siBRCA2 (40 nM, 48 h; 5′-UUGACUGAGGCUUGCUCAGUUdTdT-3′);
siTRIM25 (50 nM, 72 h; 5′-GGCUCAGAACACUUGAUAUTT-3′);
siHERC5 (50 nM, 72 h, 5′-GGACUAGACAAUCAGAAAGUUdTdt-3′);
siTOP1 (50 nM, 72 h; 5′-GGAUUUCCGAUUGAAUGAUUCUCAUTT-3′)
MEFs were transfected with a mix of the following siRNAs:
siBrca2#1 (60 nM, 48 h; 5′-UGUUAGGAGAUUCAUCUGGdTdT-3′);
siBrca2#2 (60 nM, 48 h; 5′-GGCCUAGUCUCAAGAACUCdTdT-3′);
siBrca2#3 (60 nM, 48 h; 5′-GGAAUUGUAAGGUAGGCUCdTdT-3′);
mESC were transfected with the following siRNAs from Horizon Discovery:
siIsg15 (25 nM, Cat. L-167630-00-0005); siUba7 (25 nM, Cat. L-040733-01-0005);
siTrim25 (25 nM, Cat. L-065539-01-0005).

## RNA extraction and cDNA preparation

Total RNA was extracted from cell pellets using TRIzol™ Reagent (Thermo Fisher; 15596026) according to the manufacturer's protocol. RNA concentration was determined using a Thermo Scientific Nano-Drop One. The total RNA was next reverse-transcribed to cDNA by M-MLV reverse transcriptase (Promega; M1701). 0.5 mg/mL Oligo(dT)$_{15}$ primers were added to 200 ng of RNA and the mixture was incubated at 70 °C for 5 min. After 5 min on ice, M-MLV reverse-

transcriptase master mix consisting of 5× Promega buffer, dNTP mix and M-MLV reverse transcriptase was added and incubated at 40 °C for 30 min to synthesize the cDNA followed by 30 min incubation at 50 °C to enhance synthesis from RNA with secondary structures. Finally, reverse transcriptase was inactivated by incubating at 70 °C for 15 min.

## qPCR

For qPCR, either the LightCycler® SYBR Green system (Roche; 04707516001) or TaqMan® gene expression assays (Thermo Fisher Scientific) were used. LightCycler® SYBR Green system: cDNA (11.42 ng) was mixed with SYBR Green Master mix and 1 mM forward and reverse primers. The qPCR was run on a LightCycler® 480 II (Roche). TaqMan® gene expression assays: cDNA (17.13 ng) was mixed with PrecisionPLUS qPCR Master Mix (Primer Design; Z-PPLUS-5ML) and TaqMan® primers. The qPCR was run on a LightCycler® 480 II (Roche). Target mRNA abundance was calculated relative to house-keeping genes. The list of oligos used is in the Supplementary Information file as Supplementary Table 2.

## Drugs and treatments

HU (Sigma-Aldrich; H8627) was dissolved in ddH$_2$O at concentration of a 0.1 M (7.6 mg/mL) and dissolved in growth medium to a final concentration of 4 mM. Mirin (Sigma-Aldrich; M9948) was dissolved in DMSO at a concentration of 50 mM and dissolved in growth medium to a final concentration of 50 µM. Recombinant human IFNβ (Pepro-Tech; 300-02BC) was diluted in growth medium to a concentration of 1000 U/mL and dissolved in growth medium to a final concentration of 30 U/mL. Mouse IFNβ (Sigma-Aldrich; I9032-1VL) was diluted in growth medium to a concentration of 25'600 U/mL and was dissolved in growth medium to a final concentration of 30 U/mL. Etoposide (Sigma-Aldrich; E1383) was dissolved in DMSO at a concentration of 10 mM and dissolved in growth medium to a final concentration of 5 µM. Olaparib (Selleckchem; AZD2281) was dissolved in DMSO at a concentration of 20 mM and dissolved in growth medium to a final concentration of 10 µM. Cisplatin (Sigma-Aldrich; 232120) was dissolved in PBS at a concentration of 10 mM and dissolved in growth medium to a final indicated concentration.

## Western blotting

Cells were lysed in RIPA buffer (50 mM Tris-HCl pH 7.4, 150 mM NaCl, 1% Triton X-100, 1% sodium deoxycholate, 0.1% SDS) or NP40 buffer (100 mM Tris-HCl pH 7.4, 300 mM NaCl, 2% NP40) supplemented with the following inhibitors: 50 mM sodium fluoride, 20 mM sodium pyrophosphate, 1 mM sodium orthovanadate, 1 mM phenylmethylsulfonyl fluoride (PMSF) and 1× Protease Inhibitor Cocktail (Sigma-Aldrich; P8340) for 10 min on ice. For total extracts, cells were treated with preheated (95 °C) 1% SDS (10 mM Tris-HCl, pH 8.0; 1% SDS; 1 mM sodium orthovanadate; 1× Protease Inhibitor Cocktail (Sigma-Aldrich; P8340)) and incubated for 10 min at 95 °C. The lysates were sonicated by Bioruptor (Diagenode) at 4 °C on the highest setting for 10 min (30 s on and 30 s off cycles) and centrifuged at $18,500 \times g$ for 10 min. Protein concentration was measured by Bradford protein assay (Bio-Rad; 5000006). Lysates were diluted with Laemmli Buffer (60 mM Tris-HCl pH 6.8, 2% SDS, 10% glycerol, 5% β-mercaptoethanol, 0.01% bromophenol blue), boiled at 95 °C for 3 min or at 55 °C for 10 min when BRCA2 was detected, equal amounts were loaded to polyacrylamide gels and ran at 160 V at room temperature. Proteins were blotted for 80 min (350 mA, room temperature) onto Amersham Protran 0.2 µm nitrocellulose membranes (GE Health-care). Membranes were blocked in 5% milk in TBS-Tween for at least 2 h and incubated with primary antibodies overnight at 4 °C. Following three washes in TBS-Tween, secondary antibodies were added for 1 h at room temperature. Membranes were washed three times in TBS-Tween and detected with WesternBright ECL HRP substrate (Advansta; K-12045-D50).

## Antibody generation

Antibodies against human ISG15 and TRIM25 were raised by immunizing rabbits with the recombinant His-tagged proteins expressed in *E. coli* and purified with Ni-NTA (Qiagen) and containing the full-length ISG15 and the amino acids 100–450 of TRIM25. In the case of ISG15, the serum was purified with the following procedure: 200–300 mg of the antigen was loaded onto SDS–PAGE, then transferred to a nitrocellulose membrane and then stained with Ponceau S. The area of the membrane containing the antigen was cut out, blocked with 2% BSA in TBS-T for 1 h and then incubated with the 4 mL of serum overnight at 4 °C. Bound antibodies were eluted with 0.15 M glycine-HCl, pH 2.3. 1 M Tris-HCl, pH 8.8, was immediately added to neutralize the pH of the antibody solution to pH 7.5.

## Immunofluorescence

Following siRNA transfection and IFNβ treatment, cells were grown on sterile 13-mm diameter glass coverslips. 48 h after siRNA transfection, cells were treated with 5 μM etoposide for 1 h or irradiated with 4 Gray on a Faxitron as indicated. Cells were washed with PBS, fixed with 4% paraformaldehyde for 10 min and permeabilized with 0.3% Triton X-100 in PBS for 5 min. Cells were blocked in 5% BSA in PBS for 1 h followed by incubation with primary antibodies for 1 h at room temperature. The following primary antibodies and dilutions were used: γH2AX (Millipore; 05-635; 1:800), RAD51 (Bioacademia; 70-001; 1:2000). Cells were washed with PBS and incubated with secondary antibodies for 30 min at room temperature. Total DNA was stained with 4′,6-diamidino-2-phenylidole (DAPI, 1 μM) for 5 min at room temperature. Images were acquired on a Leica DM6 microscope.

## Quantitative image-based cytometry (QIBC)

Images were acquired using Leica DM6 microscope under nonsaturating conditions and identical settings were applied to all samples within one experiment. Images were then converted to file system suitable for analysis with Olympus ScanR Image analysis software (version 3.0.1). Nuclei segmentation was performed using DAPI signal, and further RAD51 foci detection performed using integrated spot detection module. The quantification of RAD51 foci were exported and analyzed using Spotfire data visualization software (TIBCO, version 7.0.1). Number of replicates are indicated in the figure legends.

## Embryonic stem cell viability assay

PL2F7 cells were used for cell viability rescue experiments in *Brca2* conditional knockout ESCs. Viability assay was performed as described in ref. 35. In brief, 20 μg of PGK-Cre plasmid DNA were electroporated into 1×10⁷ mESCs suspended in 0.9 mL of PBS by Gene Pulser (Bio-Rad) at 230 V, 500 mF. HAT selection was started 36 h after electroporation and lasted for 5 days, followed by selection in HT media for 2 days and then normal M15 media until colonies became visible. Colonies were picked into 96-well plate. For extracting genomic DNA, colonies were lysed in 50 mL mESC buffer (10 mM Tris-HCl, pH 7.4), 10 mM EDTA, 10 mM NaCl, 5 mg/mL sodium lauroyl sarcosinate, 1 mg/mL proteinase K at 55 °C overnight, and DNA was precipitated by 100 mL 75 mM NaCl in absolute ethanol. Genomic DNA was rinsed by 70% ethanol and digested by EcoRV at 37 °C overnight for Southern blot.

## Southern blot

*Eco*RV-digested DNA was electrophoresed on 1% agarose gel in 1× TBE (0.1 M Tris, 0.1 M boric acid and 2 mM EDTA, pH 8.0) and transferred to nylon membrane. DNA probe to distinguish conditional *Brca2* allele (*Brca2*-Flox, 4.8 kb) and *Brca2* knockout allele (*Brca2*-KO, 2.2 kb) was labeled by [a-32P]-dCTP by Prime-It II Random Primer Labeling Kit (Agilent Technologies) and hybridized with Hybond-N nylon membrane (GE Healthcare) at 65 °C overnight. Membrane was washed twice with saline sodium citrate phosphate (SSCP) buffer containing 0.1%

SDS and exposed in phosphor image screen overnight and subsequently developed in Typhoon image scanner.

## Chromatin extraction

SUM149PT EV and ISG15 WT cells were plated (3×10⁶) in 15 cm plates in triplicates for each experimental condition and treated with 1 μg/mL of doxycycline. Forty-eight hours after doxycycline treatment, cells were optionally treated with HU (4 mM, 4 h). Cells were collected in ice-cold harvesting buffer (10 mM NEM, 1 mM PMSF, 1× protease inhibitor cocktail). Small fractions of cells (1/20) were lysed in 1% SDS 50 mM Tris-HCl, pH 8 (95 °C for 10 min) and loaded as total cell extracts. The remaining cells were subjected to cell fractionation by resuspending the cell pellet in ice-cold buffer A (10 mM HEPES pH 7.5, 50 mM NaCl, 300 mM sucrose, 0.5% Triton-X, 1× protease inhibitor cocktail, 1 mM PMSF, 10 mM NEM, 10 μM PJ-34). After centrifugation, supernatants were kept as cytosolic fraction. The pellet was further treated with ice-cold buffer B (10 mM HEPES pH 7, 200 mM NaCl, 0.5% NP-40, 1 mM EDTA, 1× protease inhibitor cocktail, 1 mM PMSF, 10 mM NEM, 10 μM PJ-34). After centrifugation, supernatants were kept as nuclear soluble fraction. Pellets were further treated with ice-cold buffer C (10 mM HEPES pH 7, 500 mM NaCl, 1% NP-40, 1 mM EDTA, 1× protease inhibitor cocktail, 1 mM PMSF, 10 mM NEM, 10 μM PJ-34) and sonicated for 10 min with high intensity and 30 sec on/off cycles. After centrifugation, supernatants were subjected to mass spectrometry analysis. Protein concentration was measured by Bradford method according to manufacturer's protocol.

## Mass spectrometry analysis

**Sample preparation.** Protein samples were denatured using 4% SDS at 95 °C for 10 min followed by reduction with 2 mM tris-2-carboxyethyl-phosphine and alkylation with 15 mM Chloroacetamide for 30 min at 30 °C in the dark. Single-Pot Solid-Phase-enhanced Sample-Preparation (SP3) was used for sample clean-up prior to tryptic digestion as described previously[70]. On beads trypsin digestion was performed overnight at 37 °C at a trypsin:protein ratio of 1:100 in 50 mM triethylammonium bicarbonate. The reaction was quenched with 1% trifluoroacetic acid followed by samples desalting using Stage-Tip C18 columns. Samples were dried on SpeedVac and dissolved in 3% acetonitrile/0.1% formic acid for further nanoLC-MS/MS analysis.

**NanoLC-MS/MS analysis.** Mass spectrometry analysis was performed on an Orbitrap Fusion Lumos mass spectrometer (Thermo Scientific) coupled to ACQUITY UPLC M-Class System (Waters). Biological triplicate samples were acquired in a randomized order to allow label free quantitation analysis. Loaded peptides were trapped on an ACQUITY UPLC M-Class Symmetry C18 Trap column and eluted on an ACQUITY UPLC M-Class HSS T3 column (Waters). Peptides were separated and eluted with a 90-min gradient of 35% acetonitrile/ 0.1% formic acid at a flow rate of 300 nL/min.

Data acquisition was performed using data-dependent operation mode. Full-scan MS spectra (300–2000 *m/z*) were acquired at a resolution of 120,000 at 200 *m/z* using Easy Spray Ion Source with spray voltage set to 2.3 kV. MS/MS data were acquired using higher energy collision dissociation (HCD) fragmentation.

**Data analysis.** Raw data were searched by Mascot search engine (Matrix Science) against the human proteome database (UniProt entry 9606, taxonomy, 20190709), using cysteine carbamidomethylation as a fixed protein modification. Variable modifications consisted of methylation, oxidation, acetylation, deamidation and di-glycine addition on lysine residues. Precursor mass tolerance was set to 10 ppm and a maximum of two missed cleavages was allowed. Raw data were converted to Mascot Generic Format (MGF) using Proteome

Discoverer, v1.4 (Thermo Fisher Scientific, Bremen, Germany) using the automated rule based converter control[71]. Data processing was performed using Scaffold software (version 5.1, Proteome Software Inc., Portland, OR, USA). Protein identifications were accepted if they scored over 95% probability. Protein and peptide thresholds were set at 1% and 0.1% FDR, respectively, and a minimum number of two identified peptides for each protein was allowed.

The mass spectrometry proteomics data were handled using the local data management system B-Fabric[72] and all relevant data have been deposited to the ProteomeXchange Consortium via the PRIDE[73] partner repository with the dataset identifier PXD045154.

**Statistical analysis.** Fisher's exact test implemented in Scaffold software was used to demonstrate a statistically significant association between two sample categories, based on total spectrum count of individual proteins, with $p < 0.05$.

#### TOP1 ISGylation
HEK293T cells were plated ($2 \times 10^6$) in 10 cm plates 24 h prior to transfection with plasmids encoding indicated proteins. Forty-eight hours after transfection, cells were harvested in ice-cold PBS, pelleted by centrifugation and lysed in ice-cold lysis buffer (50 mM Tris-HCl, pH 7.4, 150 mM NaCl, 1% Triton-X100, 0.1% SDS, 1% sodium deoxycholate, 10 mM NEM, 10 mM sodium pyrophosphate, 50 mM sodium fluoride, 1 mM PMSF, protease inhibitor cocktail, P8340). Following 10 min of incubation on ice, lysates were sonicated for 15 min using high intensity with 30 sec on/off cycles and centrifuged for 20 min at $16,000 \times g$ at 4 °C. Protein quantification was performed using Bradford method. Equal amounts of the supernatants were incubated with equilibrated anti-FLAG M2 magnetic beads (Sigma-Aldrich M8823) for 2 h at 4 °C while rotating. Resin was washed 4X in lysis buffer and proteins eluted in elution buffer (0.5 mg/mL of 3xFLAG-peptide (Sigma-Aldrich; F4799), 50 mM Tris HCl pH 7.4, 150 mM NaCl) for 30 min at room temperature. Supernatants were collected as FLAG-immunopurified fraction and analyzed by Western blot.

#### Cell proliferation and cell death assays
$BRCA1^{-/-}$ and $BRCA1^{+/+}$ MDA-MB-436 cells ($3 \times 10^5$) were seeded in 6-cm plates 24 h prior to siRNA transfection (50 nM of siLuc or siISG15). After 24 h, cells were trypsinized and cells ($1 \times 10^4$) were seeded in 12-well plate. Cell Titer Blue assay (Promega; G8080) was performed to assess cell viability according to manufacturer's protocol. Relative proliferation rate was quantified and normalized to day 1 (48 h post siRNA transfection).

For the cell death assays, $BRCA1^{-/-}$ MDA-MB-436 cells ($3 \times 10^5$) were seeded in 6-cm plates 24 h prior to siRNA transfection (50 nM of siLuc or siISG15). After 72 or 120 h, cells were collected by trypsinization. Using $7 \times 10^5$ cells per conditions, Annexin V and PI staining was performed using eBioscience™ Annexin V Apoptosis Detection Kit FITC (Invitrogen; 88-8005-74) according to manufacturer's protocol. In short, cells were washed once in PBS, then once in Binding Buffer. Cells were resuspended in Binding Buffer and Annexin V FITC was added; cells were incubated for 15 min at room temperature and subsequently washed in Binding Buffer. Cells were resuspended in Binding Buffer and PI staining solution was added. Annexin V and PI staining were acquired on an Attune NxT Flow Cytometer (Thermo Fisher Scientific) and analyzed using FlowJo software V.10.7.2 (FlowJo). For compensation, single-stained samples were used.

#### FACS
BRCA2-proficient ($Brca2^{+/+}$ $p53^{-/-}$) and BRCA2-deficient ($Brca2^{-/-}$ $p53^{-/-}$) KB2P cell lines derived from mouse mammary tumor were transduced by lentiviral transduction. Post transduction, cells were selected with blasticidin (10 μg/mL) for 10 days and then EV and ISG15 expression were induced by doxycycline (48 h, 2 μg/mL). Transduced cells carrying the GFP construct were sorted by FACS (BD FACSAria™ III Cell Sorter) with a 488 nm argon ion laser based on their GFP fluorescence (using BD FACSDiva 9.0.1 software). Sorted GFP positive cells were kept in culture for 1 week and then used for clonogenic assay.

#### Clonogenic assay
EV and ISG15 were induced by doxycycline (48 h, 2 μg/mL) in pCW57.1 GFP and pCW57.1 MYC-ISG15 WT-GFP in the $Brca2^{+/+}$ and $Brca2^{-/-}$ mouse KB2P transduced cell lines. GFP-positive cells were sorted by FACS and used for colony survival assay. Cells were seeded in 6-cm plates at low density with and without doxycycline (2 μg/mL). Twenty-four hours after seeding, cells were treated with Cisplatin at different concentrations for 6 h. Drug treated medium was washed out and cells were allowed to grow in a complete growth medium for 7 days in the presence of doxycycline. The colonies detected were fixed, stained with Brilliant Blue R (Sigma-Aldrich; B0149) and subsequently analyzed with the Gel-counter by Oxford Optronix and appertaining Software (version 1.1.2.0). The survival was plotted after combining 3 independent experiments as the mean surviving percentage of colonies after drug treatment compared to the mean surviving colonies from the non-treated samples.

#### Statistics and reproducibility
Number of biological replicates is defined in the legends of the figures. Results were analyzed using GraphPad, using Kruskal–Wallis (for experiments with >2 conditions) or Mann–Whitney test (experiments with 2 conditions), (two-tailed $P$ value; $P$ value > 0.05 was considered not significant, ns). The RAD51 foci counts were extracted from the raw data and subjected to statistical analysis using GraphPad Prism 9 (two-tailed $P$ value). The results were analyzed using Spotfire and GraphPad Prism9 using Kruskal–Wallis test. Significance of enrichment of proteins in mass spectrometry experiment was analyzed using Scaffold software (version 5.1, Proteome Software Inc., Portland, OR, USA) using Fisher's exact test with a significance level of $P < 0.05$. In cell proliferation assay, absorbance of each sample (technical triplicate) was normalized on untreated samples. Figure 4b, d were independently repeated twice with similar results. Figure 4e was performed once. Figure 5d, e were independently repeated three times with similar results. Figure 6a, b, d were independently repeated three times, with similar results. All statistical test results are listed in the Source data under Summary statistics.

#### Reporting summary
Further information on research design is available in the Nature Portfolio Reporting Summary linked to this article.

### Data availability
The mass spectrometry proteomics data have been deposited to the ProteomeXchange Consortium via the PRIDE[73] partner repository with the dataset identifier PXD045154. Data plotted in the mass spectrometry graph are presented in the Supplementary Data 1. Further information and requests for reagents and resources should be directed to the corresponding author. Source data are provided with this paper.

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

## Acknowledgements

We thank Sven Rottenberg, Dong-Er Zhang, Klaus-Peter Knobeloch, Satoshi Inoue, Yilun Liu, and Neil Johnson for sharing reagents. We thank the Functional Genomics Center of the University of Zurich (FGCZ) for services and Franziska Walser for technical assistance. This work was supported by the Intramural Research Program, Center for Cancer Research, National Cancer Institute, US National Institutes of Health to S.K.S.; by NWO VIDI grant (VI.Vidi.193.131) and Dutch Cancer Society (KWF-11008) to A.R.C.; by the Spanish Agencia Estatal de Investigación (PID2019-109222RB- I00/AEI/10.13039/501100011033), co-funded by European Union Regional Funds (FEDER) to R.F.; and by the Swiss Cancer Research foundation (KFS-4577-08-2018), the Worldwide Cancer Research (22-0181) and the Swiss National Science Foundation (SNSF; grant 310030_184966) to L.P.

## Author contributions

R.N.M. and U.B. performed the DNA fiber experiments, the immuno-fluorescence and proliferation studies; F.D.D. and R.N.M. performed the biochemical and functional studies on TOP1, respectively; M.S. performed the mass spectrometry analysis; M.C.R. made the initial observation; S.S.K. and S.K.S. designed and performed the viability studies in mESCs; P.D. and A.R.C. performed the drug sensitivity studies in mouse KB2P cells; R.F. developed the ISG15 and TRIM25 antibodies; L.P. conceived the project, designed the experiments, and wrote the manuscript, supported by R.N.M. and U.B.

## Competing interests

The authors declare no competing interests.
