## [Peer Review File · Nature Communications]

Interferon restores replication fork stability and cell viability in BRCA-defective cells via ISG15REVIEWER COMMENTS

Reviewer #1 (Remarks to the Author):

Biswas et al. have undertaken the study of IFN-mediated rescue of replication fork stress in the context of BRCA1/2 mutations. More specifically, IFN is able to rescue DNA damage following loss of BRCA1/2. Despite the induction of hundreds of ISGs, they pinpoint the effect of this rescue through the activity of the ubiquitin-like protein ISG15. Specifically, the effect on replication fork elongation requires ISG15 conjugation mediated by the E3 enzyme TRIM25. The finding is very novel and impactful with potential implications for new therapeutic avenues but more specific mechanistic studies about what ISG15 substrates are required for this effect and how the lack of ISG15 leads to cell death would be appropriate for publication in this journal and would extend the authors' findings beyond their previous published studies as well as that of other groups. This reviewer suggests the following experiments to address these concerns:

Major points:

The finding that ISG15 deletion is synergistic with BRCA2 deletion is compelling, as is the data implicating TRIM25, however the paper lacks mechanism of how this contributes to cell death.

- The authors could map which ISGylated targets at the replication fork account for this effect?
- Alternatively, the authors could test known ISGylated targets as identified in Wardlaw et al. 2022; PMID: 36216822

Furthermore, while ISG15 depletion affects BRCA2 deficient cell survival, the pathways or mechanisms of cell death are not explored or defined.

- How are the cells undergoing cell death?
- How does ISG15 stabilize replication forks mechanistically in this context?
- Is the effect through activity of other complementary DNA-repair pathways?

In order to further characterize the effect of ISGylation on this process:

- What do the full ISG15 blots look like - i.e. is ISGylation under these conditions restricted to a few substrates? If so, which substrates?
- Why is the HERC5 negative data not shown? Has it been previously published?
- Since HERC5 is the dominant IFN-induced ISG15 E3 - how does TRIM25 outcompete HERC5 following IFN treatment? Would this be through localization or relative induction?
- Could the TRIM25 mediated effect be through its activity as a DNA damage induced ubiquitin E3? Do the pathways synergize? Have the authors observed ubiquitin and ISG15 in whole cell lysates upon depletion of TRIM25? Could they observe ISG15 or ubiquitin levels at the replication forks?

Minor points:

Page 4:

Line 3 - Please say more about what has been found in the introduction (less vague descriptions of published work) and how this work makes a conceptual advance from the other studies.

Line 16 - did the authors show that the effect of ISGylation on BRCA2 was through fork instability?

For many of the studies the authors use siRNA for BRCA2 - is this sufficient? Would CRISPR/deletion or inducible deletion be more appropriate?

Is BRCA2 ever directly or indirectly IFN-induced (looks upregulated by IFNB in U2OS) and could its upregulation in the siRNA cells following treatment be the primary driver of restored fork ratios of 1?

Typographical errors

Page 3:

Line 23 - Some of the citations are in a different format than the rest

Line 28 - same thing - two different citation styles

Line 29 - "replication stress" not replications stress

Page 4:

Line 29 - synthesized not "synthetized"

Reviewer #2 (Remarks to the Author):

In this study, Biswas et al. show that induction of Type I IFN β can suppress the replication fork protection defect observed in BRCA1/2-defective cancer cells. This is achieved by the ubiquitin-like modifier ISG15, whereby overexpression of ISG15 is necessary and sufficient for fork protection rescue. Moreover, ISG15 overexpression leads to resistance of BRCA2 defective cells to cisplatin. Overall, the study provides an interesting study connecting inflammatory signaling to fork stability. The author may want to consider the following points

- 1) The observation that overexpression of ISG15 can rescue fork protection is interesting but lacks mechanistic or molecular understanding how this is achieved. Petrini et al showed that ISG15 is recruited to stalled forks and controls MRE11, which in turn of course controls fork degradation. Is the observed phenotype a mere function of MRE11? If so it may not be as surprising given Petrini's previous work, although an important extension. Is ISG15 epistatic with MRE11 for the rescue?
- 2) Both cGAS and STING have shown to have direct protein functions at forks with seemingly opposing functions (Siderova for STING and, Li Lan for cGAS not requiring enzymatic activity) – is this a type I interferon response, a cGAS or a STING dependent response, or all of the above
- 3) The authors say ISG15 is overexpressed in BRCA2 defective cancer cells: if so, why do these still show a fork protection defect?
- 4) ISG15 overexpression is not comparable in Fig 4 causing the relative greater difference between BRCA2 defective and proficient cells in survival, e.g. may not be BRCA2 specific and ISG15 may cause a similar difference in survival in WT cells when cisplatin concentrations are increased (hence the needed controls for BRCA2 proficient cells and replication effects of ISG15 overexpression/knockdown in the earlier figures)
- 5) The study exclusively uses siRNA to show BRCA2 defects. Does the correlation hold true in cancer cells with hypomorphic BRCA mutations?
Statistic analysis is inconsistent within the manuscript (e.g. different statistical test for fiber assays)

and sometimes obscure.

Fig 2 is missing controls without BRCA2 knockdown

In line 13 change from "BRCA2-deficient" to "BRCA2-depleted" as it is knockdown of BRCA2

In lines 13-14 it mentions that "fork stabilization in BRCA2-deficient cells is reversed by loss of ISG15, UBE1L and TRIM25" however in Figure S1g-S1i no knockdown of ISG15 is shown along with UBE1L and TRIM25.

Keep the statistical analysis consistent. With fiber data the test that is used is the Mann-Whitney nonparametric t test.

For RAD51 foci analysis the authors need to look at more than one time point to draw their conclusions. HR-dependent RAD51 foci are formed at least 4 hours after treatment The authors need to look at RAD51 foci within 4 to 8 hours post etoposide or radiation treatment.

In the introduction inconsistency with citations formatting. Most are depicted with numbers, but few are with name and year – e.g. line 23.

Reviewer #3 (Remarks to the Author):

Biswas et al "Interferon restores replication fork stability...."

This is an interesting and well-written manuscript dealing with the issue of drug resistance of BRCA1/2 deficient breast cancers, and also the mechanisms by which such cancer cells proliferate despite having high levels of DNA replication-associated genomic instability. Based on the results presented, the authors hypothesize that replication forks are stabilized in the following manner: 1) The fork damage triggers the innate immune response, causing production of interferon beta (IFN β); 2) IFN β stimulates transcription of many genes including ISG15, an ubiquitin-like modifier; 3) ISG15 stabilizes replication forks by "ISGylation" of targets that are (presumably) involved in replication fork repair or progression.

While the results are enticing and consistent with their hypothesis, the paper falls short in nailing down the hypothesis convincingly.

In particular:

1) Key evidence to support the hypothesis is that IFN β activation of ISG15 "rescues" DNA replication in mutant/knockdown cells are the DNA fiber analyses presented in Figs 1 and 2. They conclude that IFN β decreases fork degradation after HU exposure. However, it can't be ruled out that IFN β decreases fork stalling instead. The authors should have examined potential fork asymmetry to rule out such a possibility. They only measured IdU/CldU ratio unidirectionally.

2) The data in support of the "fork stabilization via ISGylation" mechanism aren't totally convincing. The evidence is genetic, using depletion of the E2 and E3 ubiquitination enzymes. However, alternative mechanisms can't be ruled out. For example, there is evidence that TRIM25 regulates p53 (<https://doi.org/10.1038/onc.2015.21>); thus, the cell cycle checkpoint pathway may underlie the observed results. Furthermore, the authors didn't show, in any of their singly mutant cells, how components of the ISG15-ubiquitination pathway may affect replication fork stability. What are potential substrates/targets?

3) The authors included different pathological and normal cell lines to test the universality of their hypothesis. However, the mESC data doesn't seem to support their hypothesis. In Fig S3c, fork stability in BRCA2-deficient mESCs doesn't seem to change when treated with IFN, unless cells were stressed by HU. It is possible IFN enables survival of BRCA2 cells via mechanisms other than fork protection.

4) The model in Fig. 4f essentially proposes that the high genomic instability causes low genomic instability. If elevated ISG15 suppresses replication stress, then wouldn't cytoplasmic DNA go down? It seems that for this general model to be correct, it would have to invoke a wave-like function or something of that sort in which genomic instability in a cell or its progeny goes up and down. If so, how would that be consistent with the experimental results?

Minor points:

- Why is "foci" (of RAD51) italicized in several places? Does it imply something other than a spot of antibody localization?
- End of Results section concerning RAD51 foci could use a final sentence summarizing the potential meaning of the results.
- Regarding ISG15 upregulation conferring chemo-resistance to KB2P BRCA2- cells: does this assume these were not already resistant to cisplatin? A key experiment would have been to test cells that were already CP or PARP1-inhibitor resistant.

POINT-BY-POINT RESPONSE TO REVIEWERS' COMMENTS

Reviewer #1 (Remarks to the Author):

Biswas et al. have undertaken the study of IFN-mediated rescue of replication fork stress in the context of BRCA1/2 mutations. More specifically, IFN is able to rescue DNA damage following loss of BRCA1/2. Despite the induction of hundreds of ISGs, they pinpoint the effect of this rescue through the activity of the ubiquitin-like protein ISG15. Specifically, the effect on replication fork elongation requires ISG15 conjugation mediated by the E3 enzyme TRIM25. The finding is very novel and impactful with potential implications for new therapeutic avenues but more specific mechanistic studies about what ISG15 substrates are required for this effect and how the lack of ISG15 leads to cell death would be appropriate for publication in this journal and would extend the authors' findings beyond their previous published studies as well as that of other groups. This reviewer suggests the following experiments to address these concerns:

We thank the reviewer for the appreciation of our work and the valuable suggestions, which contributed to improve the manuscript and strengthen our findings. To address the reviewer's comments and suggestions, we performed many additional experiments, which are now included in the revised version.

Major points:

The finding that ISG15 deletion is synergistic with BRCA2 deletion is compelling, as is the data implicating TRIM25, however the paper lacks mechanism of how this contributes to cell death.

R: Prompted by the reviewers' suggestions, we have now performed more experiments and added new data, which I am confident will strengthen our study.

- The authors could map which ISGylated targets at the replication fork account for this effect?
- Alternatively, the authors could test known ISGylated targets as identified in Wardlaw et al. 2022; PMID: 36216822

R: This is indeed a key point that we have now addressed by performing a complete set of new experiments, which are presented in the revised version (NEW Fig. 5). Over the past 20 years, many different screenings have been conducted to identify ISG15 targets, including the one indicated by the reviewer (Wardlaw et al, 2022), using different conditions to promote high levels of ISGylation, including overexpression, treatment with high doses of IFN, or infection by pathogens, resulting in about 2,000 potential ISG15 targets. This supports the notion that ISG15 is involved in many different functions and highlights the promiscuity of the system, which makes identification of specific targets and sites rather challenging. This is made even more complicated by the fact that the conditions of induction we used in all the experiments (DNA fiber assays, survival experiments of mESCs, immunofluorescence to measure RAD51 foci formation, etc.) are very mild and do not induce detectable formation of ISG15 conjugates, as shown in the NEW Supplementary Fig 4a, b, thus making the identification of ISG15 targets via the canonical strategy (pull-down of ISGylated proteins in stringent conditions) even more arduous.

Therefore, to shed light on the mechanism underpinning the effect of IFN β /ISG15 on the stability of the replication fork, we chose to take a different approach and assess – by mass spectrometry analysis – how the chromatin composition varies under the conditions in which ISG15 promotes fork stability and cross-reference this list of proteins with potential ISGylated factors reported in the literature, as suggested by the reviewer.

At this point, to avoid the IFN β treatment that might induce perturbations in protein expression and chromatin composition that are unrelated to the effect on DNA replication, we favored the use of the engineered SUM149PT cell lines expressing MYC-ISG15 or the empty vector and performed the experiment in biological triplicates for statistical significance, by comparing cells expressing MYC-ISG15 or the empty vector (EV), following doxycycline induction and optional treatment with HU. Our analysis identified more than 2000 proteins, of which 191 appeared differently regulated upon ISG15 induction. Proteins with a fold change of less than ± 0.5 were not considered further. The remaining factors – 72 upregulated and 62 downregulated – were then cross-referenced with the list of putative ISG15 substrates identified. Among them we found TOP1, which appeared particularly interesting since it was identified as potential ISG15 targets in three different reports (Zhang, 2019; Pinto-Fernandez, 2012; Wardlaw, 2022) and for its crucial role in replication dynamics and genome integrity. We then proceeded with the functional validation of TOP1 and could observe that, in condition of over-expression of the ISGylation machinery, it does undergo ISGylation (NEW Fig. 5d, e). Importantly, the DNA fiber assay revealed that the IFN β -mediated restoration of fork protection in BRCA1/2-deficient cells completely relies on TOP1, since its depletion results in the degradation of newly synthesized DNA at similar extent as in BRCA1/2-deficient cells (NEW Fig. 5f, g). The new data included in Fig. 5 of the revised manuscript provide important mechanistic insights into the IFN β /ISG15-mediated restoration of fork protection, which are discussed in the revised manuscript and now included in our working model (Fig. 6f).

Furthermore, while ISG15 depletion affects BRCA2 deficient cell survival, the pathways or mechanisms of cell death are not explored or defined.

•How are the cells undergoing cell death?

R: Indeed, we have observed that depletion of ISG15 in MDA-MB-436 cells reduces cell survival (previous Fig. 4C). To address the point raised by the reviewer, we have analyzed the cell death in this system by FACS. As included in the NEW Supplementary Fig. 5a, we performed FACS analysis at day 3 and 5 upon siISG15 or siLuc transfected cells, and measure cell death by using either PI (to assess necrotic cells) and Annexin V (to assess apoptotic cells). As positive control, we treated cells at 55°C for 20 min and then mixed with untreated cells. We observed that at day 5 after ISG15 depletion the majority of cells are apoptotic, while in siLuc-transfected cells up to 84% of cells are viable.

•How does ISG15 stabilize replication forks mechanistically in this context?
•Is the effect through activity of other complementary DNA-repair pathways?

R: We thank this reviewer for this important remark, which has prompted us – also based on new data obtained during revision – to formulate a more defined mechanistic model for ISG15-dependent stabilization of stalled forks in BRCA-defective cells. In a nutshell, we envision that ISG15-mediated acceleration of fork progression counteracts replication fork reversal, in favor of RECQ1-mediated fork restart (Raso et al., JCB 2020). As fork degradation in BRCA-defective cells reportedly requires reversed forks as entry points for deregulated nucleolytic activities (Taglialatela, 2017; Mijc, 2017; Lemacon, 2017; Kolinjivadi, 2017), ISG15-mediated fork restart would rescue fork integrity in BRCA-defective cells by counteracting reversed fork accumulation. Importantly, in the course of this revision, we have found that ISG15-mediated rescue of fork integrity in BRCA-defective cells strictly requires Topoisomerase-I (TOP1). We are convinced that this genetic requirement reflects the need to resolve topological constraints, in order for ISG15 to foster fork restart, thereby counteracting fork reversal and the associated degradation. This interpretation is in line with original findings that had identified fork reversal as a mechanism to redistribute topological stress at stalled forks (Postow et al., 2001), implying the need to overcome these constraints when reversed forks are being restarted. We are convinced that these new data provide novel and relevant mechanistic insight into the effect of IFN β signaling and ISG15 over-expression on fork integrity in genetic backgrounds prone to stalled fork degradation. As requested by the reviewer, we have highlighted these new mechanistic aspects in the revised manuscript.

In order to further characterize the effect of ISGylation on this process:

•What do the full ISG15 blots look like - i.e. is ISGylation under these conditions restricted to a few substrates? If so, which substrates?

R: The conditions of induction we are using throughout all the experiments (DNA fiber assays, survival experiments of mESCs, immunofluorescence to measure RAD51 foci formation, etc..) are very mild (IFN β 30 U/ml, 2 h) and do not induce detectable formation of ISG15 conjugates. For sake of clarity, we have now included examples of the full ISG15 immunoblots NEW Supplementary Fig 4a, b, and compared with conditions in which the ISGylation is detectable (as upon higher dose of IFN β treatment, e.g., 250 U/ml).

•Why is the HERC5 negative data not shown? Has it been previously published?

R: We thank the reviewer for raising this point. The DNA fiber data on HERC5 were not included in the first submission because we could not properly confirm its depletion by Western blotting, being HERC5 expressed at extremely low levels, and having inconsistent data with the qPCR on HERC5 mRNA in different replicates. Now we have adopted different kit and probes (as indicated in the Methods) and could successfully re-analyze the samples, confirming HERC5 depletion. These data are now included in the revised manuscript (NEW Supplementary Fig. 1h, j).

•Since HERC5 is the dominant IFN-induced ISG15 E3 - how does TRIM25 outcompete HERC5 following IFN treatment? Would this be through localization or relative induction?

R: The reviewer is right, HERC5 is the dominant E3 ISG15 ligase induced by IFN; in fact, in unstimulated cells, HERC5 protein and mRNA levels are extremely low and usually not detectable by Western blot and barely detectable by qPCR. Even treatment with low dose of IFN β (30 U/mL), as adopted throughout the manuscript, is not sufficient to detectably induce HERC5 by Western blot, while we could observe a slight increase by qPCR, but far lower compared to high dose of IFN (as shown in the 'Data for reviewers only', Result 1). On the other side, TRIM25 is constitutively expressed and its protein levels ready detectable by Western blotting, as shown throughout the manuscript. Therefore, we believe that TRIM25 is favored over HERC5 because it is much more abundant, and it is reportedly present both in the nucleus and cytoplasm.

•Could the TRIM25 mediated effect be through its activity as a DNA damage induced ubiquitin E3? Do the pathways synergize? Have the authors observed ubiquitin and ISG15 in whole cell lysates upon depletion of TRIM25? Could they observe ISG15 or ubiquitin levels at the replication forks?

R: We cannot completely rule out the possibility that TRIM25 acts on DNA replication via additional mechanisms, which might include its ubiquitin ligase activity. As suggested by the reviewer, we tested the levels of ubiquitin and ISG15 in TRIM25 wild type and knockout cells; in total cell extracts – as well as in different cellular fractions, including chromatin fractions – we could not detect significant differences. These data are now included in the 'Data for reviewers only', Result 2.

Minor points:

Page 4:

Line 3 - Please say more about what has been found in the introduction (less vague descriptions of published work) and how this work makes a conceptual advance from the other studies.

R: We have now added a new paragraph in the introduction that better explains the reported role of ISG15 in DDR and replication and how our study brings conceptual advances.

Line 16 - did the authors show that the effect of ISGylation on BRCA2 was through fork instability?

R: We have shown the ISGylation is required for fork stability in different contexts and for the viability of Brca2-deficient mESCs. We did not test directly if it is also required for the increased drug resistance in KB2P cells. We have now rephrased this part to make our conclusions more precise.

For many of the studies the authors use siRNA for BRCA2 - is this sufficient? Would CRISPR/deletion or inducible deletion be more appropriate?

R: Throughout our studies we have used several different systems stably carrying mutations or deletion of BRCA genes – as detailed below – to prove the key points of the study centered on the role of IFN β /ISG15 in the BRCA contexts: restoration of fork protection, restoration of viability in mES cells, acquisition of drug resistance. In particular, we made use of cells carrying Brca2 deletion: KB2P and PL2F7 cell lines that were engineered to obtain Brca2 deletion by means of the Cre-loxP system (Jonkers et al., Nat Genetics 2001; Kuznetsov et al, Nat Medicine 2008). We also used cells carrying hypomorphic BRCA mutations: SUM149PT triple-negative breast tumour cells possess a commonly occurring hypomorphic BRCA1 exon 11 c.2288delT frameshift mutation and loss of the WT BRCA1 allele, and MDA-MB-436 had the 5396 + 1G>A mutation in the splice donor site of exon 20. In both cell lines these mutations were accompanied by loss of the other BRCA1 allele (Elstrodt et al, Cancer Res, 2006). Moreover, we also made use of PL2F7 cells expressing BRCA2 R2336H variant, PL2F7-Brca2^{-/-};BRCA2(R2336H), which only express the hypomorphic allele (Biswas et al., 2011; Kuznetsov et al., 2008).

Is BRCA2 ever directly or indirectly IFN-induced (looks upregulated by IFNB in U20S) and could its upregulation in the siRNA cells following treatment be the primary driver of restored fork ratios of 1?

R: We also noticed that in some experiments the levels of BRCA2 depletion looks rather different. We believe this is due to the problematic detection of BRCA2 by Western blot, due to the high MW and protein instability. We addressed this concern by assessing the levels of BRCA2 mRNA upon siRNA following optional treatment with IFN β by qPCR and we obtained very similar measurements. A representative analysis is now included in the revised manuscript (NEW Supplementary Fig. 1a). Moreover, since the same effect was observed in several different systems carrying a stable loss or mutations in BRCA1 and BRCA2, we tend to exclude that the effect we measured in U2OS is due to the induction of BRCA2 expression.

Typographical errors

Page 3:

Line 23 - Some of the citations are in a different format than the rest

R: We apologize for this inconvenience and have now fixed them.

Line 28 - same thing - two different citation styles

R: We now fixed them.

Line 29 - “replication stress” not replications stress

R: We now fixed it.

Page 4:

Line 29 - synthesized not “synthetized”

R: We now fixed it.

Reviewer #2 (Remarks to the Author):

In this study, Biswas et al. show that induction of Type I IFN β can suppress the replication fork protection defect observed in BRCA1/2-defective cancer cells. This is achieved by the ubiquitin-like modifier ISG15, whereby overexpression of ISG15 is necessary and sufficient for fork protection rescue. Moreover, ISG15 overexpression leads to resistnace of BRCA2 defective cells to cisplatin.

Overall, the study provides an interesting study connecting inflammatory signaling to fork stability. The author may want to consider the following points

We are glad that the reviewer considers our study interesting and thankful for the precious suggestions and comments. We hope that the additional data included in the revised manuscript help to further strengthen our findings.

1) The observation that overexpression of OSG15 can rescue fork protection is interesting but lacks mechanistic or molecular understanding how this is achieved. Petrini et al showed that ISG15 is recruited to stalled forks and controls MRE11, which in turn of course controls fork degradation. Is the observed phenotype a mere function of MRE11? If so it may not be as surprising given Petrini’s previous work, although an important extension. Is ISG15 epistatic with MRE11 for the rescue?

R: In the revised manuscript, we present a complete set of new data that add more mechanistic insight on how ISG15 upregulation promotes fork stability (NEW Fig. 5). We have performed a mass spectrometry-based analysis to investigate how chromatin composition varies under conditions in which ISG15 promotes fork stability. At this point, to avoid the IFN β treatment that might induce perturbations in protein expression and chromatin composition that are unrelated to the effect on DNA replication, we favored the use of the engineered SUM149PT cell lines expressing Myc-ISG15 or the empty vector and performed the experiment in biological triplicates for statistical significance, by comparing cells expressing Myc-ISG15 or the empty vector (EV), following doxycycline induction and optional treatment with HU. Our analysis identified more than 2000 proteins, of which 191 appeared differently regulated upon ISG15 induction. Proteins with a fold change of less than \pm 0.5 were not considered further. The remaining factors – 72 upregulated and 62 downregulated – were then cross-referenced with the list of putative ISG15 substrates identified. Among them we found TOP1, which appeared particularly interesting since it was identified as potential ISG15 targets in three different reports (Zhang, 2019; Pinto-Fernandez, 2012; Wardlaw, 2022) and for its crucial role in replication dynamics and genome integrity. We then proceeded with the functional validation of TOP1 and could observe that, in condition of over-expression of the ISGylation machinery, it undergoes ISGylation (NEW Fig. 5d, e). Importantly, the DNA fiber assay revealed that the IFN β -mediated restoration of fork protection in BRCA1/2-deficient cells completely relies on TOP1, since its depletion results in the degradation of newly synthesized DNA at similar extent as in BRCA1/2-deficient cells (NEW Fig. 5f, g). The new data included in Fig. 5 of the revised manuscript provide important mechanistic insights into the IFN β /ISG15-mediated restoration of fork protection, which are discussed in the revised manuscript and now included in our working model (Fig. 6f).

2) Both cGAS and STING have shown to have direct protein functions at forks with seemingly opposing functions (Siderova for STING and, Li Lan for cGAS not requiring enzymatic activity) – is this a type I interferon response, a cGAS or a STING dependent response, or all of the above

R: The observation that both STING and cGAS regulate DNA replication, although with likely opposite effects, is indeed rather interesting. Both studies investigated the effect of loss of these factors (i.e., STING or cGAS), by

transient depletion or genetic ablation, directly on DNA replication fork progression and stability, without the activation of the type I IFN response.

In Lazarchuk et al (Front Mol Biosci 2023), the proposed model suggests that the inner nuclear membrane pool of STING is activatable by cGAS and other unidentified putative factors, and it is directly responsible for its effect on DNA replication for stability, by promoting nucleolytic degradation of the stalled forks, without the intervention of the canonical activation via IRF3 and the induction of IFN signaling.

In the cGAS study (Chen et al, Sci Adv 2020), the authors aimed to investigate a nuclear function of cGAS independently of its role in immune response and of STING itself. Hence, they did not stimulate the type I IFN response with either IFN-stimulating DNAs or directly by IFN treatment. They focused on the effect of loss of cGAS (by CRISPR/Cas9 gene editing) and complementation with different cGAS mutants on cell proliferation and DNA replication. Therefore, the effect they observed is largely, if not exclusively, independent on the type I IFN response.

In our cases, the effect in DNA replication fork stability is observed upon mild dose of IFN β (30 U/ml, 2 hr) but also upon over-expression of the sole ISG15 (i.e., without IFN β treatment), therefore suggesting that neither cGAS nor STING are involved.

3) The authors say ISG15 is overexpressed in BRCA2 defective cancer cells: if so, why do these still show a fork protection defect?

R: We are grateful to the reviewer for raising this point. We believe it is a matter of dose. BRCA2-deficient cells show increased levels of ISG15 compared to BRCA2-proficient cells. This increase in ISG15 levels is moderate compared to that obtained by IFN stimulation (even at the low dose we used, 30 U/mL, 2 h) and likely not sufficient to restore detectable fork stability. We believe that for ISG15 to exert its effect on fork stability, a greater induction is required, which may be achieved with time upon BRCA defects and progressively contribute to chemoresistance.

4) ISG15 overexpression is not comparable in Fig 4 causing the relative greater difference between BRCA2 defective and proficient cells in survival, e.g. may not be BRCA2 specific and ISG15 may cause a similar difference in survival in WT cells when cisplatin concentrations are increased (hence the needed controls for BRCA2 proficient cells and replication effects of ISG15 overexpression/knockdown in the earlier figures)

R: The reviewer is right, we observed variations in the expression of MYC-ISG15 in different experiments, but they go in both directions, as shown in Data for reviewers only”, Result 3. However, despite this difference, the effect of ISG15 upregulation on drug response was very similar in both cases, probably because upregulation was still much higher than endogenous levels, being sufficient to exert its effects on fork protection and drug response.

5) The study exclusively uses siRNA to show BRCA2 defects. Does the correlation hold true in cancer cells with hypomorphic BRCA mutations?

R: Throughout our studies we have used several different systems stably carrying mutations or deletion of BRCA genes – as detailed below – to prove the key points of the study centered on the role of IFN β /ISG15 in the BRCA contexts: restoration of fork protection, restoration of viability in mES cells, acquisition of drug resistance. We have now put more emphasis on this aspect in the revised manuscript.

We made use of cells carrying Brca2 deletion: KB2P and PL2F7 cell lines that were engineered to obtain Brca2 deletion by means of the Cre-loxP system (Jonkers et al., Nat Genetics 2001; Kuznetsov et al, Nat Medicine 2008). We also used cells carrying hypomorphic BRCA mutations: SUM149PT triple-negative breast tumour cells possess a commonly occurring hypomorphic BRCA1 exon 11 c.2288delT frameshift mutation and loss of the WT BRCA1 allele, and MDA-MB-436 had the 5396 + 1G>A mutation in the splice donor site of exon 20. In both cell lines these mutations were accompanied by loss of the other BRCA1 allele (Elstrodt et al, Cancer Res, 2006). Moreover, we also made use of PL2F7 cells expressing BRCA2 R2336H variant, PL2F7-Brca2^{-/-};BRCA2(R2336H), which only express the hypomorphic allele (Biswas et al., 2011; Kuznetsov et al., 2008).

Statistic analysis is inconsistent within the manuscript (e.g. different statistical test for fiber assays) and sometimes obscure.

[few rows below] Keep the statistical analysis consistent. With fiber data the test that is used is the Mann-Whitney nonparametric t test.

R: We used the Mann-Whitney nonparametric t-test in the fiber experiments in which we compared two conditions, while when comparing more conditions, the use of a nonparametric ANOVA, i.e., the Kruskal-Wallis test, is recommended. The major difference between the Mann-Whitney U and the Kruskal-Wallis H is simply that the latter can accommodate more than two groups. Both tests require independent (between-subjects) designs and use summed rank scores to determine the results. Examples can be found in: Zhu et al, Nat Comms 2022

(Shao lab); Stoy et al, NSMB 2023 (Lopes lab); Cong et al, Mol Cell 2021 (Cantor lab); Fletcher et al, J Clin Invest 2023 (Coleman lab); Liu et al, Science 2023 (Cortez lab).

Fig 2 is missing controls without BRCA2 knockdown

R: We thank this reviewer for this important remark. We have now included new results showing the effect of the sole depletion of ISG15 and ISGylating enzymes on fork stability. Interestingly, we observed that loss of ISG15 or UBE1L per se leads to the instability of newly synthesized DNA upon fork stalling, highlighting an important effect of ISG15 and ISGylation in the protection of stalled forks. Moreover, in support of these findings, we now show that in context of ISG15 depletion (i.e., degradation of stalled forks), the re-expression of the conjugation-defective mutant (dGG, missing the C-terminal GlyGly motif that forms covalent bound with targets) fails to restore fork protection. These important results are now included in the revised manuscript (NEW Fig. 1, NEW Fig. 3b, f, g).

In line 13 change from “BRCA2-deficient” to “BRCA2-depleted” as it is knockdown of BRCA2

R: We now changed it accordingly to the suggestion and fixed similar mistakes throughout the manuscript.

In lines 13-14 it mentions that “fork stabilization in BRCA2-deficient cells is reversed by loss of ISG15, UBE1L and TRIM25” however in Figure S1g-S1i no knockdown of ISG15 is shown along with UBE1L and TRIM25.

R: We now changed the sentence to avoid confusion.

For RAD51 foci analysis the authors need to look at more than one time point to draw their conclusions. HR-dependent RAD51 foci are formed at least 4 hours after treatment. The authors need to look at RAD51 foci within 4 to 8 hours post etoposide or radiation treatment.

R: We have now repeated the analysis of RAD51 foci 4 hr after etoposide treatment in U2OS cells upon depletion of BRCA1 or BRCA2 and obtained similar results as upon after 1 hr, showing a partial contribution of IFN β to the formation of RAD51 foci upon BRCA depletion. These data are now included in the 'Data for reviewers only', Result 4.

In the introduction inconsistency with citations formatting. Most are depicted with numbers, but few are with name and year – e.g. line 23.

R: We apologize for this inconvenience and have now fixed them.

Reviewer #3 (Remarks to the Author):

Biswas et al “Interferon restores replication fork stability....”

This is an interesting and well-written manuscript dealing with the issue of drug resistance of BRCA1/2 deficient breast cancers, and also the mechanisms by which such cancer cells proliferate despite having high levels of DNA replication-associated genomic instability. Based on the results presented, the authors hypothesize that replication forks are stabilized in the following manner: 1) The fork damage triggers the innate immune response, causing production of interferon beta (IFN β); 2) IFN β stimulates transcription of many genes including ISG15, an ubiquitin-like modifier; 3) ISG15 stabilizes replication forks by “ISGylation” of targets that are (presumably) involved in replication fork repair or progression.

While the results are enticing and consistent with their hypothesis, the paper falls short in nailing down the hypothesis convincingly.

In particular:

1) Key evidence to support the hypothesis is that IFN β activation of ISG15 “rescues” DNA replication in mutant/knockdown cells are the DNA fiber analyses presented in Figs 1 and 2. They conclude that IFN β decreases fork degradation after HU exposure. However, it can't be ruled out that IFN β decreases fork stalling instead. The authors should have examined potential fork asymmetry to rule out such a possibility. They only

measured IdU/CldU ratio unidirectionally.

R: We are thankful to the reviewer for this observation. In these experiments fork stalling is induced by depletion of dNTPs via high dose of HU treatment (4 mM, 4 h) that inhibits the ribonucleotide reductase. Therefore, the forks are stalled because no dNTPs are available. The intention of this set of experiments is to test whether the stalled forks are protected, i.e., no extensive degradation is observed. To directly address the concern arisen by the reviewer and clarify this point, we performed a dedicated DNA fiber assay in SUM149PT cells, in which we added hydroxyurea (HU) during the second labelling (IdU) and compared replication fork progression in cells pre-treated with IFN β (as we do in all our experiments, i.e., 30 U/mL for 2 h, followed by 46 h chasing) and untreated cells. In both cases, we found a very similar effect on fork stalling. This result is now included in the 'Data for reviewers only', Result 5A. Moreover, we included observation that we have previously reported (Raso et al, 2020), in which we showed that the upregulation of ISG15 in U2OS Flp-In T-REx does not significantly impact fork asymmetry 'Data for reviewers only', Result 5B, C. Taken together these results indicate that IFN β does not decrease fork stalling induced by high dose of HU treatment.

2) The data in support of the “fork stabilization via ISGylation” mechanism aren't totally convincing. The evidence is genetic, using depletion of the E2 and E3 ubiquitination enzymes. However, alternative mechanisms can't be ruled out. For example, there is evidence that TRIM25 regulates p53 (<https://doi.org/10.1038/onc.2015.21>); thus, the cell cycle checkpoint pathway may underlie the observed results.

R: The reviewer is right, we have used genetic evidence – i.e., the depletion of the ISG15-specific E1 activating enzyme and of the ligases TRIM25 and HERC5, reported to promote both ubiquitination and ISGylation – to demonstrate the role of ISGylation in fork protection. This reviewer pointed out that TRIM25 can also regulate p53, which may affect the results of our study. The status of p53 in the cells employed in this study is summarized below.

U2OS: p53 wild type

SUM149PT: p53 mutated, increased expression

CAPAN-1: p53 mutated, inactive protein

MDA-MB-436: p53 is undetectable due to frameshift truncating mutation

KB2P: p53 deletion, inactive protein

Sources: Jonkers et al., Nat Genetics 2001; <https://link.springer.com/article/10.1007/s10549-006-9186-z>;

<https://www.ncbi.nlm.nih.gov/pmc/articles/PMC4694938/>; <https://www.ncbi.nlm.nih.gov/pmc/articles/PMC2860631/>

While we recognize the correctness of this observation, it should be remarked that we obtain very similar results in all the cell lines tested, regardless the p53 status.

Furthermore, the authors didn't show, in any of their singly mutant cells, how components of the ISG15-ubiquitination pathway may affect replication fork stability. What are potential substrates/targets?

This is indeed a key question, which has prompted us to perform many more experiments during revision to investigate the role of ISG15 and ISGylating enzymes in DNA replication fork protection and to formulate a more defined mechanistic model for ISG15-dependent stabilization of stalled forks in BRCA-defective cells.

We have now included new results showing the effect of the sole depletion of ISG15 and ISGylating enzymes on fork stability. Interestingly, we observed that loss of ISG15 or UBE1L per se leads to the instability of newly synthesized DNA upon fork stalling, highlighting an important effect of ISG15 and ISGylation in the protection of stalled forks. Moreover, in support of these findings, we now show that in context of ISG15 depletion (i.e., degradation of stalled forks), the re-expression of the conjugation-defective mutant (dGG, missing the C-terminal diGly motif that forms covalent bound with targets) fails to restore fork protection. These important results are now included in the revised manuscript (NEW Fig. 1, NEW Fig. 3b, f, g).

Moreover, we have performed a mass spectrometry-based analysis to investigate how chromatin composition varies under conditions in which ISG15 promotes fork stability. At this point, to avoid the IFN β treatment that might induce perturbations in protein expression and chromatin composition that are unrelated to the effect on DNA replication, we favored the use of the engineered SUM149PT cell lines expressing Myc-ISG15 or the empty vector and performed the experiment in biological triplicates for statistical significance, by comparing cells expressing Myc-ISG15 or the empty vector (EV), following doxycycline induction and optional treatment with HU. Our analysis identified more than 2000 proteins, of which 191 appeared differently regulated upon ISG15 induction. Proteins with a fold change of less than ± 0.5 were not considered further. The remaining factors – 72 upregulated and 62 downregulated – were then cross-referenced with the list of putative ISG15 substrates identified. Among them we found TOP1, which appeared particularly interesting since it was identified as potential ISG15 targets in three different reports (Zhang, 2019; Pinto-Fernandez, 2012; Wardlaw, 2022) and for its crucial role in replication dynamics and genome integrity. We then proceeded with the functional validation of TOP1 and could observe that, in condition of over-expression of the ISGylation machinery, it undergoes ISGylation (NEW Fig. 5d, e). Importantly, the DNA fiber assay revealed that the IFN β -mediated restoration of fork protection in BRCA1/2-deficient cells completely relies on TOP1, since its depletion results in the degradation of newly synthesized DNA at similar extent as in BRCA1/2-deficient cells (NEW Fig. 5f, g). The new data included in Fig. 5 of the revised manuscript provide important mechanistic insights into the IFN β /ISG15-mediated restoration of fork protection, which are discussed in the revised manuscript and now included in our working model (Fig. 6f).

3) The authors included different pathological and normal cell lines to test the universality of their hypothesis. However, the mESC data doesn't seem to support their hypothesis. In Fig S3c, fork stability in BRCA2-deficient mESCs doesn't seem to change when treated with IFN, unless cells were stressed by HU. It is possible IFN enables survival of BRCA2 cells via mechanisms other than fork protection.

R: We apologize with the reviewer for having caused confusion in this figure. The protocol adopted to measure the degradation/stability of newly synthesized DNA upon fork stalling includes the critical step of HU-treatment, to deplete dNTPs and force the cells to stop replication. Hence, in most of the experiments we have performed throughout the study, the experimental pipeline adopted is the same, as reported in NEW Fig. 1d, although the HU treatment is not indicated in each figure but only in the figure legends. In the figure to which the reviewer refers (NEW Fig. Supplementary 3c), we have indicated HU treatment as opposed to the DMSO samples.

4) The model in Fig. 4f essentially proposes that the high genomic instability causes low genomic instability. If elevated ISG15 suppresses replication stress, then wouldn't cytoplasmic DNA go down? It seems that for this general model to be correct, it would have to invoke a wave-like function or something of that sort in which genomic instability in a cell or its progeny goes up and down. If so, how would that be consistent with the experimental results?

R: Our interpretation of the data included in this study is that the elevated levels of ISG15 observed in BRCA contexts do not suppress replication stress, but rather 'help' BRCA-deficient cancer cells to proliferate and duplicate their DNA, regardless the DNA repair defects and exaggerated replication fork degradation, fostering genome instability. Hence, ISG15 is important for the fitness of cancer cells, since its depletion results into marked reduction of cell proliferation and increased apoptosis (as revealed by the new experiment performed during revision to address additional reviewers' concerns and now included in the NEW Supplementary Fig. 5a. Thanking the important input and suggestions received by the reviewers, the new data obtained during this revision work allowed us to propose a more defined hypothesis, included in the Discussion, explaining the role of ISG15 in replication fork dynamics and stability.

Minor points:

- Why is "foci" (of RAD51) italicized in several places? Does it imply something other than a spot of antibody localization?

R: We used italic since we refer to the Latin word, but we can omit since it is the same in English.

- End of Results section concerning RAD51 foci could use a final sentence summarizing the potential meaning of the results.

R: We now have added a sentence summarizing the take-home message, as suggested.

- Regarding ISG15 upregulation conferring chemo-resistance to KB2P BRCA2- cells: does this assume these were not already resistant to cisplatin? A key experiment would have been to test cells that were already CP or PARP1-inhibitor resistant.

R: Yes, these cells are quite sensitive to cisplatin, and this is the reason why we have used them to test our hypothesis. The suggestion of this reviewer is very appropriate and in fact we are planning future studies to investigate the effect of loss of ISG15 (and its conjugation) on the survival and drug response of BRCA1/2-deficient cell lines that are considered resistant to therapy.

Result 1

qPCR showing levels of HERC5 mRNA in different conditions

mRNA levels of HERC5 are very low in untreated (UT) condition as well as upon low dose (30 U/mL) of IFN treatment, while it greatly increase upon higher dose of IFN (250 U/mL). Representative samples obtained from cells transfected with different siRNAs are shown.

Result 2

Levels of ubiquitinated and ISGylated proteins in parental vs TRIM25 knockout HEK293T cells

Cells were subjected fractionation to isolate different subcellular fractions, which were resolved by SDS-PAGE (8 and 12%), together with the total cell lysates (TCL).

Result 3**MYC-ISG15**

Protein levels of MYC-ISG15 in BRCA2-proficient (*Brca2*^{+/+} *p53*^{-/-}) and -deficient (*Brca2*^{-/-} *p53*^{-/-}) KB2P cells (relative to Fig. 6d). Two representative experiments are shown.

Result 4

RAD51 foci in BRCA1/2-depleted cells upon IFN

Following siRNA transfection and IFN β treatment, cells were grown on sterile 13-mm diameter glass coverslips. 48 h after siRNA transfection, cells were treated with 5 μ M etoposide for 1 h, washed with PBS and released for 4 h before fixation with 4% PFA and immunostaining.

Result 5**A EFFECT OF IFN ON FORK STALLING**

SUM149PT cells were treated with IFN (30 U/mL, 2 h), followed by 46 h chasing, or left untreated. Cells were incubated with halogenated nucleotides and treated during the 2nd labelling with high dose of HU treatment (4 mM, 4 h) to induce fork stalling.

SUM149PT untreated

SUM149PT +IFNβ

Result 6

SISTER FORK ASYMMETRY

U2OS Flp-In T-REx cells (empty vector and FLAG-ISG15), induction with 1ug/ml doxy for 48 h.

Cells were optionally treated during the 2nd labelling with low dose of CPT (50 nM) to induce mild replication stress and increase fork asymmetry.

B. Each fork is described by the length of left and right IdU tracks. Orange lines define a range of 30% difference between left and right tracks; left > right + 30% and right > left + 30% are considered asymmetric. **C.** Graph summarizing the percentage of asymmetric forks as in B.

REVIEWER COMMENTS

Reviewer #1 (Remarks to the Author):

I am satisfied with the authors' revised manuscript and think that it is much improved. In particular, the addition of mass spectrometry data revealed a potential new mechanism behind the authors' previous observations.

Reviewer #2 (Remarks to the Author):

The authors changed the original manuscript quite substantially and added quite a lot new data. Unfortunately, these added data present additional challenges: While the authors added new data in Fig 5, they did not address the basic questions if the ISG15 results are dependent on MRE11, which should be addressed given Petrini's results to put their data in the context of what is already known in the field. Moreover, Fig. 1 now starts by directly comparing at ISG15 KO with WT cells. The authors measure the ratio of IdU/CldU tracts. This analysis can be used in principle under conditions with HU or replication stalling following the double label as a measure of fork degradation. However, The authors see substantially IdU shortened tracts also with non-HU treated cells (Fig. 1a). This suggests assymmetric forks under unperturbed conditions. Thus the lower IdU/CldU ratio with HU could similarly simply be a consequence of fork asymmetry, not degradation. It therefore is essential for the authors to 1) express fork degradation by comparing absolute tract lengths (before HU) with and without HU and 2) show that the shortened tracts are rescued by. Inhibition of a nuclease (MRE11, DNA2, EXO1, MUS81, EXO5 are all nucleases shown to degrade forks under given circumstances). As is, the data does not imply fork degradation but the fork data could be a mere result of replication dynamic changes. Given the new data it is essential to show that the authors observe a true nucleolytic degradation of forks (kinetically and with nuclease KO/KD/inhibition).

While overall a commendable addition, this concern extends to the interpretation of the added masspec data. First, comparing overexpression of ISG15 with basal expression of ISG15, is not the same as comparing basal expression to deletion or knock-down of the protein (whereby the authors compare these . Particularly in the context of fork stability, these reactions may cause opposing rather than similar effects

The claim that the observed tract shortening phenotypes are strictly dependent on TOP1 is an over interpretation and not well controlled for: 1) statistical comparison between TOP1si and TOP1si+IFN β are missing in Fig. 5f (assuming the missing label is TOP1si in this Figure?) and by eye looks slightly increased, suggesting the possibility of parallel pathways. If TOP1 truly is in the same pathway and assuming ISG15 defects do cause fork degradation, both should be rescued by inhibition/KO/KD of the nuclease causing the phenotype.

2) the interpretation is based on limited fork data in SUM149PT cells, which have a strong intrinsic fork degradation or tract shortening phenotype (can't tell the difference without kinetics and/or nuclease controls) presumably due to BRCA1 defects. Does IFN β treatment alone increase tract lengths in WT cells (cells without intrinsic tract shortening)? This is an important control in the interpretation and context of all results as it provides additional distinction between fork dynamics and degradation.

Moreover, TOP1si alone appears to cause a tract shortening: there is a noticeable tract shortening in Fig 5g with siTOP1 alone in U2OS cells. However, the authors seemingly selectively chose not to include significance in this graph. These data further substantiate the possibility that TOP1 and BRCA1/2 act in parallel tract shortening pathways. double knock BRCA +TOP1 and single knock down with and without MRE11i could address epistasis between the proteins. Tract shortening to the similar extend is not a measure of epistasis.

The response of the authors to point 3 is inconsistent with their interpretation: The authors say ISG15

is overexpressed in BRCA2 defective cancer cells, but think this is insufficient to restore fork protection. Yet, they claim that this overexpression is required or needed in BRCA defective cancer cells for viability. If this is correct, then tract shortening in BRCA defective cells is not correlated to sensitivity or resistance. Hence the correlation the authors are trying to make between cellular outcome and fork dynamics are just that, correlations but not necessarily causative.

Statistical analysis remains inconsistent with missing comparisons throughout the manuscript.

Reviewer #3 (Remarks to the Author):

The authors did a great job responding to my comments as well as those of the other two reviewers. I am satisfied with the revisions and have no additional concerns.

REVIEWER COMMENTS

Reviewer #1 (Remarks to the Author):

I am satisfied with the authors' revised manuscript and think that it is much improved. In particular, the addition of mass spectrometry data revealed a potential new mechanism behind the authors' previous observations.

We are grateful to this reviewer for appreciating our revision work and for supporting the publication of our revised study.

Reviewer #2 (Remarks to the Author):

The authors changed the original manuscript quite substantially and added quite a lot new data. Unfortunately, these added data present additional challenges:

We note that all data we added in our revised manuscript had been explicitly requested by the reviewers (particularly this reviewer), as perceived as important controls for our key findings. We acknowledge that some of these data open interesting new questions (on the impact of defective ISGylation on replication fork integrity) for future investigation. However, this is not directly related to the conditions of IFN signalling and ISG15 overexpression at the core of our results (which was the main reason we initially avoided including them). While we addressed the key new concerns of this reviewer here below, we strongly feel that – in the interest of clarity and focus of our story – we should refrain from focusing on the role of nucleases (MRE11, EXO1 etc.) upon loss of ISGylation, and keep our studies centred on the surprising contribution of IFN/ISG15 in a BRCA-defective context.

While the authors added new data in Fig 5, they did not address the basic questions if the ISG15 results are dependent on MRE11, which should be addressed given Petrini's results to put their data in the context of what is already known in the field.

The contribution of MRE11 to fork degradation in BRCA-defective contexts has been largely documented in previous studies (e.g. Ray Chaudhuri et al., Nature 2016; Mijic et al., Nat Comms 2017; Lemacon et al., Nat Comms 2017; Taglialatela et al., Mol Cell 2017; Kolinjivadi et al., Mol Cell 2017; ...). Practically all conditions rescuing fork degradation in BRCA-defective cells – whether by directly affecting the engaged nucleases, or by removing intermediates on which they act, i.e. reversed forks – were shown to affect MRE11 recruitment and/or action at stalled forks. In that respect, monitoring MRE11 recruitment or activity at stalled forks upon BRCA deficiency and IFN signalling/ISG15 overexpression would not provide any relevant mechanistic information on the novel phenomena reported in our paper. Similarly, showing that MRE11 is still the key (albeit not only, see Lemacon et al., Nat Comms 2017) nuclease mediating fork degradation upon BRCA deficiency, IFN signalling and TOP1 inactivation, seems highly redundant and unnecessary to support any of the key claims of our study. In fact, several data – either published previously by our group (Raso et al., JCB 2020) or included in this manuscript – clearly point to a role of ISG15 and ISGylation upstream of nucleolytic degradation, and rather impacting fork progression, architecture and dynamics. We have now included an additional dataset that significantly extends and supports this interpretation. In Fig. 5h,i, we now show that an independent condition previously shown to suppress fork degradation upon BRCA2 defects by accelerating fork progression and limiting fork remodelling (i.e. PARP inhibition prior to fork stalling by HU, see Mijic et al., Nat Comms 2017) requires the very same ISGylated factor identified by our MS screen upon ISG15 overexpression, i.e. TOP1. As now discussed in the text, these additional data strengthen the conclusion that accelerated fork progression upon IFN signalling and increased ISG15 levels is the key alteration of replication fork dynamics mediating

the rescue of fork integrity upon BRCA deficiency, setting this scenario clearly apart from other genetic conditions impacting fork integrity by affecting nuclease access and action at stalled replication forks.

Moreover, Fig. 1 now starts by directly comparing at ISG15 KO with WT cells. The authors measure the ratio of IdU/CldU tracts. This analysis can be used in principle under conditions with HU or replication stalling following the double label as a measure of fork degradation. However, The authors see substantially IdU shortened tracts also with non-HU treated cells (Fig. 1a). This suggests asymmetric forks under unperturbed conditions.

Thus the lower IdU/CldU ratio with HU could similarly simply be a consequence of fork asymmetry, not degradation.

It therefore is essential for the authors to 1) express fork degradation by comparing absolute tract lengths (before HU) with and without HU

Fig. 1a does not represent a classical fork degradation assay, as indeed it does not include an HU treatment and the data are displayed as absolute track length (and not as ratio). It is simply meant to convey that ISG15 KO cells display delayed fork progression and – in agreement with Wardlaw et al., Nat Comms 2022 – fork asymmetry. To clarify this point, besides the data in Fig. 1a, we have now included in a new figure (Supplementary Fig. 1a-d) the CldU track length (which, as expected is affected by ISG15 KO similarly to IdU), the IdU/CldU ratios from these experiments without HU (which, as expected is close to 1.0) and the analysis of fork asymmetry (expressed as ratio of IdU^{min}/IdU^{max}), hence setting these data (without HU) clearly apart from the fork degradation assays shown thereafter.

and 2) show that the shortened tracts are rescued by. Inhibition of a nuclease (MRE11, DNA2, EXO1, MUS81, EXO5 are all nucleases shown to degrade forks under given circumstances). As is, the data does not imply fork degradation but the fork data could be a mere result of replication dynamic changes. Given the new data it is essential to show that the authors observe a true nucleolytic degradation of forks (kinetically and with nuclease KO/KD/inhibition).

We have now included additional data in our possess, showing clearly that fork degradation observed upon ISG15 loss (KO in MEFs or siRNA downregulation in U2OS) is dependent on MRE11, as it is fully rescued by mirin (Fig. 1i-k).

Taken together, our data clarify that upon ISG15 loss fork progression is impaired even without exogenous treatment (Supplementary Fig. 1a-d), and that HU treatment induces MRE11-dependent fork degradation (Fig. 1i-k). In our view, considering that ISG15 loss is mainly meant in our manuscript as a control condition for other, more relevant findings, these additional data do provide all essential information to the readers to properly understand and interpret our key datasets.

While overall a commendable addition, this concern extends to the interpretation of the added masspec data. First, comparing overexpression of ISG15 with basal expression of ISG15, is not the same as comparing basal expression to deletion or knock-down of the protein (whereby the authors compare these). Particularly in the context of fork stability, these reactions may cause opposing rather than similar effects.

Throughout the paper, we have not compared ISG15 loss to ISG15 overexpression. Rather, we focused our key data interpretations on increased ISG15 levels (by overexpression or IFN treatment). In the case of our MS screen, we have explained in the text why we favoured a setup with ISG15 overexpression, in respect to IFN treatment: “To avoid the IFN β treatment that might induce perturbations in protein expression and chromatin composition that are unrelated to the effect on DNA replication”. Nonetheless, validation of our key target (TOP1) was then performed in the most relevant experimental condition for our manuscript, i.e., IFN treatment, where this

target fully confirmed its promise and functional relevance. We do agree that ISG15 loss and ISG15 overexpression (IFN signalling) are clearly distinct genetic conditions and will likely have opposite effects in terms of fork progression and integrity. Hence, we have not considered ISG15 loss for our MS studies on potential ISGylation targets.

The claim that the observed tract shortening phenotypes are strictly dependent on TOP1 is an over interpretation and not well controlled for:

1) statistical comparison between TOP1si and TOP1si+IFNb are missing in Fig. 5f (assuming the missing label is TOP1si in this Figure?) and by eye looks slightly increased, suggesting the possibility of parallel pathways.

We apologise for not including this comparison, which has now been included. The slight increase noticed by the reviewer is not statistically significant and, in any case, does not affect the key conclusion that TOP1 is strictly required for IFN-induced fork stabilization in BRCA-defective cells (in this case, SUM149PT). We also note that this key conclusion is strongly supported by analogous effects in U2OS cells, where efficient downregulation of TOP1 abrogated the IFN-mediated rescue of fork integrity in BRCA2-depleted cells. The full requirement of TOP1 in both cases excludes the possibility of parallel pathways suggested by this reviewer.

If TOP1 truly is in the same pathway and assuming ISG15 defects do cause fork degradation, both should be rescued by inhibition/KO/KD of the nuclease causing the phenotype.

As discussed above, we have included in our manuscript solid evidence suggesting that changes in fork dynamics – as opposed to altered nuclease activity – underlies IFN-mediated rescue of fork integrity in BRCA-defective cells. This conclusion is further supported by new data we are now including, showing that TOP1 – the ISGylation target identified in our work – is strictly required to rescue fork integrity in BRCA-defective cells not only upon IFN treatment, but also upon PARP inhibition (Fig. 5h,i), which was previously linked to accelerated restart of reversed forks. Also based on these data, testing that MRE11 is still the key nuclease involved in fork degradation in all genetic conditions of fork instability upon BRCA-defects does not seem a central point to ascertain the key conclusions of these studies, and would not reveal any direct epistatic relationships between the factors that have been investigated.

2) the interpretation is based on limited fork data in SUM149PT cells, which have a strong intrinsic fork degradation or tract shortening phenotype (can't tell the difference without kinetics and/or nuclease controls) presumably due to BRCA1 defects.

The conclusion that TOP1 is required for IFN-mediated restoration of fork integrity in BRCA defective cells is based on data in two different cell lines (SUM149PT, U2OS) and with both BRCA1 (SUM149PT) and BRCA2 defects (siBRCA2 in U2OS).

Does IFNb treatment alone increase tract lengths in WT cells (cells without intrinsic tract shortening)? This is an important control in the interpretation and context of all results as it provides additional distinction between fork dynamics and degradation.

We have already published that IFNb treatment and ISG15 over-expression accelerates fork progression in different BRCA-proficient cell lines (Raso et al., JCB 2020). It is also based on those data – and on additional data discussed above – that we suggest a change in fork dynamics (and presumably in fork remodelling) rather than nucleases activity, to be at the basis of IFN-mediated fork stabilization in BRCA-defective cells. It is important to note that multiple other defects in fork progression and remodelling (e.g. SMARCAL1/HLTF/ZRANB3 inactivation,

PARP inhibition) were previously shown to restore fork integrity upon BRCA-defects, without affecting directly the nucleolytic activities involved in fork degradation (see Mijic et al., Nat Comms 2017; Lemacon et al., Nat Comms 2017; Taglialatela et al., Mol Cell 2017; Kolinjivadi et al., Mol Cell 2017).

Moreover, TOP1si alone appears to cause a tract shortening: there is a noticeable tract shortening in Fig 5g with siTOP1 alone in U2OS cells. However, the authors seemingly selectively chose not to include significance in this graph. These data further substantiate the possibility that TOP1 and BRCA1/2 act in parallel tract shortening pathways. double knock BRCA +TOP1 and single knock down with and without MRE11i could address epistasis between the proteins. Tract shortening to the similar extend is not a measure of epistasis.

We have not directly highlighted the effect of TOP1 downregulation on fork stabilization and progression, as this has been extensively documented by Tuduri et al. (Nat Cell Biol 2009) and in follow-up papers. We note however that the effect noticed by the reviewer is only visible in the experimental condition associated with marked TOP1 downregulation in U2OS cells (Fig. 5g). However, also milder conditions of TOP1 downregulation that do not induce any detectable effect on fork progression in SUM149PT (Fig. 5f) do induce a full, comparable rescue of fork degradation in IFN-treated BRCA1-defective cells. This evidence excludes that mild, previously reported effects on fork progression of TOP1 inactivation are relevant for one of the key conclusions of our work.

The response of the authors to point 3 is inconsistent with their interpretation: The authors say ISG15 is overexpressed in BRCA2 defective cancer cells, but think this is insufficient to restore fork protection. Yet, they claim that this overexpression is required or needed in BRCA defective cancer cells for viability. If this is correct, then tract shortening in BRCA defective cells is not correlated to sensitivity or resistance. Hence the correlation the authors are trying to make between cellular outcome and fork dynamics are just that, correlations but not necessarily causative.

We think that slightly increased ISG15 levels may suffice to support proliferation of BRCA-defective cells, while a further boost of these levels by IFN treatment or other inflammatory responses may be required to drive chemoresistance in these cells. As explained in our revised Discussion, this model is supported by several key findings in our paper, extended to multiple cell lines, and seems to us a solid, albeit provocative conclusion to stimulate further studies on this clinically relevant research area.

Statistical analysis remains inconsistent with missing comparisons throughout the manuscript.

We apologise if statistical comparisons that were perceived by this reviewer as important for our claims were not included in the submitted manuscript. We have now paid attention that any key claim within the main text is supported by adequate statistical analysis in the figures.

Reviewer #3 (Remarks to the Author):

The authors did a great job responding to my comments as well as those of the other two reviewers. I am satisfied with the revisions and have no additional concerns.

We are grateful to this reviewer for appreciating our revision work – also in respect to comments of the other reviewers – and for supporting the publication of our revised study.